# NF-κB is a central regulator of hypoxia-induced gene expression

Dilem Shakir [ID][1], Michael Batie [ID][1], Chun-Sui Kwok[1], Simon J Cook[2], Niall S Kenneth [ID][1] & Sonia Rocha [ID][1]✉

## Abstract

Hypoxia is both a physiological and pathological signal in cells. Changes in gene expression play a critical role in the cellular response to hypoxia, enabling cells to adapt to reduced oxygen availability. These changes are primarily mediated by the HIF family of transcription factors, however, other transcription factors such as NF-κB, are also activated by hypoxia. Although NF-κB is known to be activated by hypoxia, the extent to which NF-κB contributes to the hypoxic response remains poorly understood. Here, we analysed hypoxia-induced, NF-κB-dependent gene expression, to define the NF-κB-dependent hypoxic signature. Our analysis reveals that most genes downregulated by hypoxia require NF-κB for their repression. We show that although the NF-κB-mediated hypoxic response may vary between cell types, a core subset of hypoxia-inducible genes requires NF-κB across multiple cell backgrounds. We demonstrate that NF-κB is critical for reactive oxygen species (ROS) generation and regulation of genes involved in oxidative phosphorylation under hypoxia. This work highlights NF-κB's central role in the hypoxia response and offering new insights into gene expression regulation by hypoxia and NF-κB.

**Keywords** Hypoxia; NF-kappaB; RNA-seq; Transcriptional Repression; IKK
**Subject Category** Chromatin, Transcription & Genomics

## Introduction

Hypoxia, or reduction in oxygen homeostasis, is both a physiological and pathological cue (Kaelin and Ratcliffe, 2008; Kenneth and Rocha, 2008; Xie and Simon, 2017). As oxygen is required for viability of many organisms, complex and intricate mechanisms have evolved for cells and organisms to be able to sense hypoxia and respond appropriately. These responses vary from increasing oxygen supply, to reducing oxygen consumption, to eventual adaptation or cell death (Rocha, 2007). At the cellular level, the best understood response occurs via the Hypoxia Inducible Factor (HIF) transcription factor family. HIFs are controlled by a family of 2-oxoglutarate-dependent dioxygenases called Prolyl-hydroxylases (PHDs), that modify HIF, making it a target for ubiquitylation via the von Hipple Lindau (VHL) containing E3-ligase complex and destruction by the 26S proteasome (An and Rettig, 2005; Bruegge et al, 2007; Fandrey et al, 2006). In addition to HIF, other transcription factors are also activated in response to hypoxia, including the NF-κB family of transcription factors (Cummins and Taylor, 2005; Kenneth and Rocha, 2008). NF-κB is best known for its role in innate immunity and its response and control of inflammation, but is also activated in response to physical, physiological and oxidative stresses (Perkins and Gilmore, 2006). The NF-κB transcription factor family is composed of 7 distinct proteins that are encoded by 5 different genes, namely RelA (p65), RelB, cRel, NF-κB1 (p105/p50) and NF-κB2 (p100/p52) (Perkins and Gilmore, 2006). Unlike the other NF-κB subunits, NF-κB1 and NF-κB2 are synthesised as pro-forms, p105 and p100, which are proteolytically processed to their respective active forms, p50 and p52 (Perkins and Gilmore, 2006). Regulation of the NF-κB family occurs at several levels from transcription to post-translational modifications and also cytoplasmic sequestration via their interaction with a family of inhibitors, called Inhibitors of κB, or IκBs (Perkins and Gilmore, 2006). Given its complexity, NF-κB gene targets are numerous and can also be characterised based on their timing of expression, such as early, mid and late genes (Tian et al, 2005). Furthermore, NF-κB activity is under several positive and negative feedback loops depending on the activating stimulus, including NF-κB subunits themselves, IκBs, and upstream activators and regulators (Prescott et al, 2021). NF-κB is activated by almost all stresses investigated so far but is best characterised following infection with bacteria or virus, or in response to pro-inflammatory cytokines recognised by members of the tumour necrosis factor receptor superfamily (TNFRSF) (Perkins, 2007). Receptor activation initiates a signalling cascade involving the IκB kinase (IKK) complex that phosphorylates IκB, leading to its degradation and the release of NF-κB for nuclear translocation and control of gene transcription (Adhikari et al, 2007). Many more levels of complexity and control exist in the NF-κB pathway, making it one of the most complex transcription factors to study. However, its importance is well established in immunity, response to inflammation, cell death, proliferation and DNA damage responses (Almaden et al, 2016; Gerondakis and Siebenlist, 2010; Liu et al, 2017; Qin et al, 2007; Volcic et al, 2012; Xia et al, 2018; Xu et al, 2015).

NF-κB activation in response to hypoxia has been observed by several studies, including some of our own (Al-Anazi et al, 2018;

[1]Department of Biochemistry, Cell and System Biology, Institute of Systems, Molecular and Integrative Biology, University of Liverpool, Liverpool L697ZB, UK. [2]Signalling Programme, The Babraham Institute, Babraham Research Campus, Cambridge CB22 3AT, UK. ✉E-mail: srocha@liverpool.ac.uk

Chandel et al, 2000; Culver et al, 2010; D'Ignazio et al, 2017; D'Ignazio and Rocha, 2016; Koong et al, 1994; Melvin et al, 2011; Patel et al, 2017; Taylor et al, 1999). While the exact mechanism of activation slightly differs from study to study, it is well established that hypoxia-dependent activation of NF-κB requires IKK activation, and this has been seen in the fruit fly, mouse and human cells (Bandarra et al, 2014; Culver et al, 2010). Additional mechanistic insights include a requirement for a calcium-dependent IKK upstream kinase (Culver et al, 2010), Transforming Growth Factor Activated Kinase (TAK1) (Adhikari et al, 2007; Melvin et al, 2011), input from PHD enzymes (Cummins and Taylor, 2005; Wilson et al, 2020) and lack of IκB degradation (Culver et al, 2010).

It is well established that cells have a specific transcriptional response to hypoxia, and analysis of several RNA-sequencing datasets revealed a conserved hypoxia inducible signature (Puente-Santamaria et al, 2022). There is also a recognised set of genes that are repressed in response to hypoxia, but the mechanisms controlling this are poorly understood (Batie et al, 2018; Cavadas et al, 2017). However, whether the NF-κB response to hypoxia is a general feature of most cells, or if there is a common gene expression signature, is not known. Here we combined analysis of publicly available datasets across different cellular backgrounds, with transcriptomics analysis of siRNA-mediated depletion of NF-κB subunits in hypoxia to uncover the contribution of NF-κB to the cellular response. Our analysis shows NF-κB is required for activation of a subset of hypoxia activated genes (i.e., 30%) but strikingly is required for the repression of around 60% of hypoxia repressed genes. Our analysis reveals a core set of hypoxia-induced NF-κB target genes across all cell types, with some cell type specificity. Interestingly, most of the genes were not previously known to be dependent on NF-κB to be regulated in hypoxia, and only a small subset is recognised NF-κB target genes, demonstrating the context-dependent nature of NF-κB. Lastly, we demonstrate that NF-κB is required to maintain reactive oxygen levels necessary for cell signalling.

## Results

### Transcriptional response profile of hypoxia stimulation is NF-κB dependent

As the majority of NF-κB target genes have been characterised in the context of inflammatory signalling, we aimed to determine the contribution NF-κB to the transcriptional response to hypoxia within a given cell type. As such, to identify hypoxia-inducible, NF-κB-dependent gene signatures, an unbiased high-throughput RNA-sequencing (RNA-seq) analysis was performed. HeLa cells were transfected with control siRNA or siRNA to NF-κB subunits, RelA, RelB or cRel and exposed to 21% $O_2$ (normoxia) or 1% $O_2$ (hypoxia) for 24 h (h) prior to profiling for global transcriptomic analysis using RNA-seq (Fig. 1A). Principal component analysis of the collected RNA-seq data showed that samples clustered by condition (Appendix Fig. S1A). Control samples treated with normoxia clustered together with relatively weak correlation to the rest of the samples, while hypoxia-treated samples grouped close to one another, with the exception of siRelA-treated samples (Appendix Fig. S1A). Also, close clustering of replicates within each condition demonstrated their relative similarity to each other.

In this analysis, efficiency of the hypoxia stimulation was investigated using DESeq2 normalised counts acquired from RNA-seq analysis showing differential expression of the genes in hypoxia samples, compared to normoxia. As expected, the expression of Carbonic Anhydrase (CA9), a core hypoxia upregulated gene, was increased in response to hypoxia (Appendix Fig. S1B). Additionally, DESeq2 normalised counts of RelA, RelB, and cRel were reduced by the respective siRNA treatments, demonstrating efficient siRNA depletion (Appendix Fig. S1C–E). Successful depletion of NF-κB subunits by siRNA treatment was shown at the protein level by Western blot (Appendix Fig. S1F). To investigate the impact of individual NF-κB subunit depletions on hypoxia gene expression changes, volcano plots were generated following differential expression analysis by comparing each condition to normoxia control (Fig. 1B–E). This analysis revealed that more than 2,000 genes were differentially expressed at 5% false discovery rate (FDR) and log2 fold change 0.58 in hypoxia control siRNA treated samples compared to normoxia control siRNA treated samples (Fig. 1B, Dataset EV1). Overall, RelA depletion induced the highest transcriptional changes, followed by RelB or cRel depletions upon hypoxia stimulation (Fig. 1C–E, Dataset EV1). When comparing hypoxia inducible gene signatures in the absence of NF-κB subunits, a total of 2611 genes were upregulated in hypoxia exposed, NF-κB subunit depleted cells, compared to normal oxygen control cells (Fig. 1F, Dataset EV1). Out of these 2611 genes, 1070 genes (41%) were shared when RelA, RelB or cRel were individually depleted (Fig. 1F, Dataset EV1). Conversely, 2188 genes were downregulated in hypoxia exposed NF-κB subunit depleted cells, compared to normal oxygen control cells, with only 295 out of the 2188 genes (13%) shared across all 3 subunits (Fig. 1G, Dataset EV1).

To identify NF-κB-dependent hypoxia-inducible gene signatures, we performed integrative analysis, comparing differentially hypoxia-regulated genes in control cells (hypoxia control compared to normoxia control) with the combined list of all NF-κB subunit depleted, differentially hypoxia-regulated genes (hypoxia siRelA/siRelB/sicRel compared to normoxia control). Genes which are differentially regulated in hypoxia (control siRNA treated cells), but not in RelA/RelB/cRel depleted cells, are considered RelA/RelB/cRel-dependent hypoxia responsive genes. This analysis revealed that 35.5% (556 out of 1568) of the hypoxia-upregulated genes (Fig. 2A; Appendix Fig. S2A), and 62.6% (410 out of 655) of the hypoxia-downregulated genes were NF-κB-dependent (Fig. 2B; Appendix Fig. S2B). Overall, this highlights the importance of NF-κB in the regulation of the transcriptional response to hypoxia. Furthermore, the heatmap generated using the NF-κB-dependent hypoxia up- and down-regulated genes showed that individual NF-κB subunit depletions displayed different transcriptional profiles (Fig. 2C). We also investigated the number of NF-κB-dependent hypoxia-regulated genes shared by the different Rel subunits, revealing that although 41 genes were simultaneous dependent on all NF-κB subunits for hypoxia upregulation, most of the other genes were regulated by individual subunits (Fig. 2D, Dataset EV2). Similar results were also obtained for repressed genes in hypoxia (Fig. 2E, Dataset EV2). This analysis revealed that the highest proportion hypoxia regulated genes are RelB-dependent followed by cRel and RelA.

To characterise the identified NF-κB-dependent, hypoxia-inducible gene signatures, their association with hallmark gene

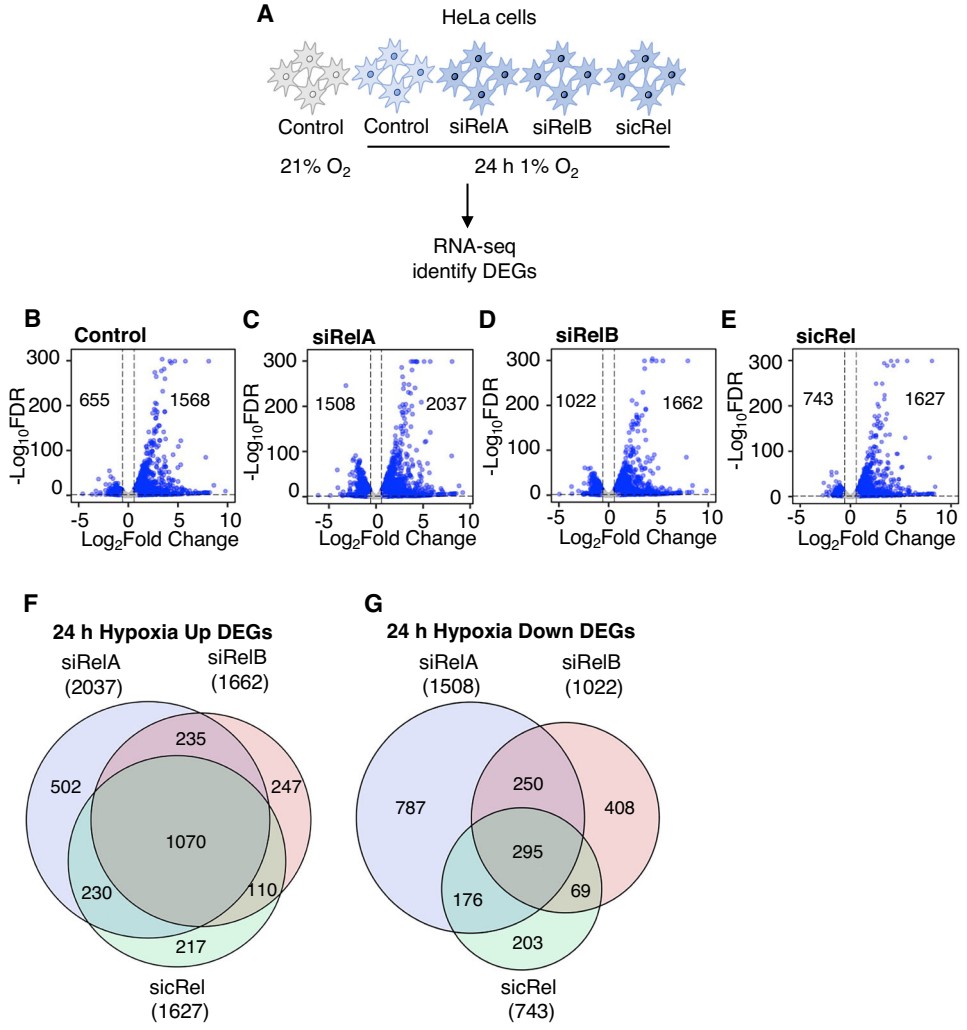

**Figure 1. Gene expression changes in response to hypoxia and NF-κB subunit depletions in HeLa cells.**

(A) RNA-seq (n = 2) in HeLa cells cultured at 21% oxygen (normoxia) or exposed to 24 h 1% oxygen (hypoxia), transfected with control siRNA or RelA, RelB or cRel siRNAs. (B–E) Differential expression analysis volcano plots for 24 h hypoxia control, siRelA, siRelB or sicRel compared to normoxia control. Blue points indicate DEGs. (F, G) Overlap of siRelA, siRelB and sicRel upregulated genes in hypoxia, compared to normoxia control (F) and siRelA, siRelB and sicRel downregulated genes in hypoxia, compared to normoxia control (G) in HeLa cells. Source data are available online for this figure.

sets was investigated. Hypoxia, glycolysis, and TNF-α signalling via NF-κB were the top three gene signatures significantly enriched at RelA-dependent 24 h hypoxia upregulated DEGs. RelB-dependent 24 h hypoxia upregulated DEGs were enriched in Epithelial–mesenchymal transition (EMT) and myogenesis gene signatures. Finally, EMT, TNF-α signalling via NF-κB, and inflammatory response were the top three gene signatures significantly enriched at cRel-dependent 24 h hypoxia upregulated DEGs (Fig. 2F). Oxidative phosphorylation and ROS gene sets were significantly enriched at RelA-, RelB- and cRel-dependent 24 h hypoxia downregulated DEGs (Fig. 2G). We next performed motif enrichment analysis in the NF-κB-dependent genes. HIF-1α and HIF-1β motifs were enriched at RelA- and RelB-dependent 24 h hypoxia upregulated DEGs (Appendix Fig. S3A), and HIF-2α and HIF-1β motifs were enriched at cRel-dependent 24 h hypoxia upregulated DEGs (Appendix Fig. S3A), suggesting a role of HIF subunits in NF-κB controlled gene expression increases in hypoxia.

The motif enrichment analysis of each NF-κB subunit-dependent hypoxia downregulated DEGs showed a variety of known NF-κB associated transcription factors (Appendix Fig. S3B), indicating distinct transcriptional regulation of RelA, RelB and cRel-dependent hypoxia downregulated gene signatures. To further elucidate the correlation of NF-κB controlled hypoxic gene expression changes with HIF binding sites, we analysed publicly available HIF subunit ChIP-seq datasets in HeLa cells exposed to hypoxia (Appendix Fig. S4A–C). Similar to motif enrichment analysis, HIF subunit (HIF-1α, HIF-2α and HIF-1β) binding sites were enriched at RelA-, RelB-, and cRel-dependent 24 h hypoxia upregulated DEGs, but not at RelA-, RelB-, and cRel-dependent 24 h hypoxia downregulated DEGs, further indicating HIF involvement in NF-κB-dependent hypoxia induced gene activation. This is not surprising given the dominant role of HIF transcription factors in hypoxia induced gene activation (Batie et al, 2022). We next investigated if NF-κB subunits had been found at the

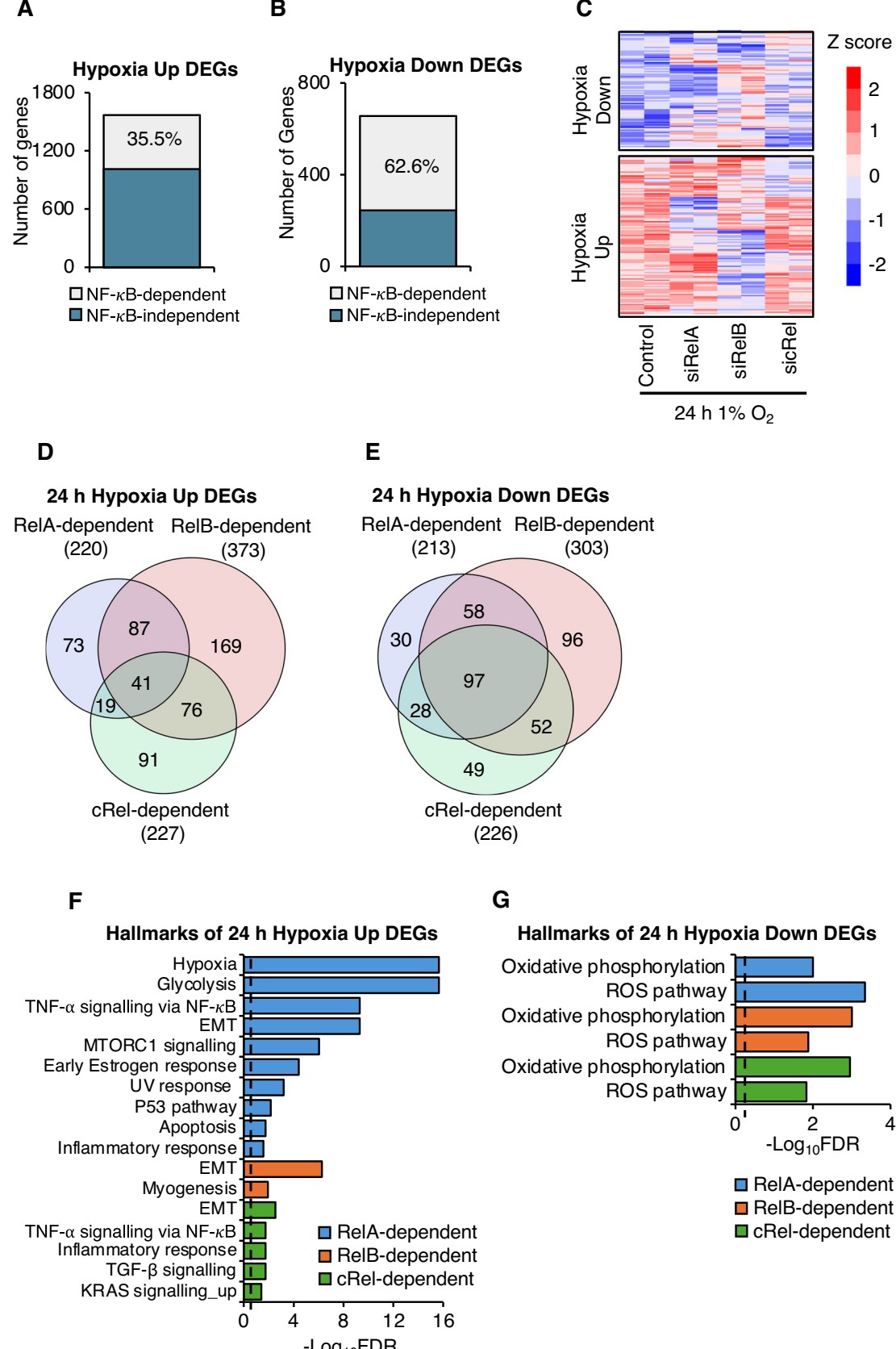

**Figure 2. NF-κB is required for 35% of hypoxia induced genes and 63% of hypoxia repressed genes.**

(A, B) NF-κB dependence of hypoxia inducible differential expressed genes (DEGs); percentage of NF-κB dependent DEGs are labelled. (C) Heatmap displaying Z score transformed gene expression levels for NF-κB dependent hypoxia regulated genes. (D, E) Overlap of RelA, RelB or cRel-dependent hypoxia up- or down-regulated DEGs. (F, G) Over representation analysis (ORA) was performed through WEB-based Gene SeT AnaLysis Toolkit using the Molecular Signatures Database hallmark gene sets, investigating NF-κB dependent hypoxia upregulated DEGs (F) or downregulated DEGs (G). Dashed line shows statistical significance threshold of FDR 0.05 (−Log$_{10}$FDR 1.3). Source data are available online for this figure.

promoters of hypoxia regulated genes. To this end, we took advantage of the ChIP-Atlas database (Zou et al, 2024). This analysis revealed significant enrichment of RelA (80%) but also RelB (22%) and cRel (18%) in both hypoxia upregulated and downregulated genes (Appendix Fig. S5A). However, similar results were obtained when analysing NF-κB subunit occupancy at NF-κB-independent hypoxia responsive genes (Appendix Fig. S5B), demonstrating enrichment of Rel binding sites from the ChIP atlas are not specific to NF-κB-dependent hypoxia responsive genes. This was expected, as many sites bound by transcription factors are not always functional, and their use varies between stimuli and cell background.

We also investigated if NF-κB-dependent, hypoxia-regulated genes have been previously described as direct NF-κB targets in the Gilmore database (Appendix Fig. S5C). Interestingly, only a small number of NF-κB target genes from this database were present, although they are favoured at NF-κB-dependent hypoxia activated over repressed genes.

In summary, this analysis showed that NF-κB plays a previously unrecognised and crucial role in the control of gene expression in hypoxia in the cell system analysed.

## NF-κB-dependent, hypoxia-inducible gene expression changes are also observed by qPCR

To validate the obtained data from the RNA-seq analysis, we selected several genes with high log2 fold change in hypoxia compared to normoxia, and some known HIF and/or NF-κB target genes amongst the 556 NF-κB-dependent, hypoxia-upregulated genes (*BCL3*, *KMT2E*, *SAP30*, *TGFA*, *EGLN3*, *USP28*, *VIM*, *NFIX*, *GADD45B*, *JUNB*, *KLF10*, *FTH1*) and 410 NF-κB-dependent, hypoxia-downregulated genes (*GCLM*, *CCND3*, *NQO1*, *AIFM1*, *SOD1*, *CASP10*, *LAMTOR2*, *IDH1*) to analyse by quantitative polymerase chain reaction (qPCR) (Fig. 3A; Appendix Table S1). HeLa cells were transfected with control siRNA or siRNAs to NF-κB subunits, RelA, RelB, or cRel and exposed, or not, to 24 h hypoxia prior to qPCR analysis. In this analysis, efficiency of the hypoxia stimulation was measured using *CA9* (Appendix Fig. S6A), as expected expression of this gene was highly upregulated in response to hypoxia. Additionally, RNA levels of *RelA*, *RelB*, and *cRel* were reduced in response to their respective siRNA treatments, demonstrating robust siRNA depletion (Appendix Fig. S6B–D). RNA levels of all the selected NF-κB-dependent, hypoxia-upregulated genes were significantly increased in hypoxia control siRNA treated samples compared to normoxia control siRNA treated samples (Fig. 3B–D). Next, effect of the NF-κB subunits was tested individually on hypoxia up- and down-regulated gene expression changes. The hypoxia-induced increase in *USP28*, *VIM*, *NFIX*, and *FTH1* was significantly decreased by RelA depletion compared to hypoxia control sample (Fig. 3B), while

*BCL3* showed a slight decrease upon RelA depletion in hypoxia, though this result was not statistically significant (Appendix Fig. S6E). *KMT2E* levels were not altered by RelA depletion (Appendix Fig. S6E). All the selected RelB-dependent hypoxia-upregulated genes, *USP28*, *VIM*, *NFIX*, *SAP30*, *TGFA*, and *EGLN3*, significantly decreased following RelB depletion in hypoxia compared to hypoxia control sample (Fig. 3C). Similarly, the hypoxia induced increase in *USP28*, *FTH1*, and *JUNB* significantly decreased following cRel depletion (Fig. 3D). *GADD45B* showed a slight decrease with cRel depletion in hypoxia, however, this result was not statistically significant (Appendix Fig. S6F) and *KLF10* was not responsive to cRel depletion (Appendix Fig. S6F). In summary, 5 out of 6 RelA-dependent hypoxia upregulated genes, 6 out of 6 RelB-dependent hypoxia-upregulated genes, and 4 out of 5 cRel-dependent hypoxia upregulated genes were successfully validated by qPCR (Fig. 3B–D). RNA levels of the selected NF-κB-dependent, hypoxia downregulated genes significantly decreased in hypoxia control siRNA treated samples compared to normoxia control siRNA treated samples (Fig. 3E,F). Following siRNA depletion of RelA, *GCLM* and *IDH1* significantly increased expression in hypoxia compared to hypoxia control (Fig. 3E). Similarly, RelB depletion followed by hypoxia exposure, increased the mRNA abundance of *GCLM*, *IDH1*, *AIFM1*, *CCND3*, and *CASP10*, compared to hypoxia control (Fig. 3F). *SOD1* slightly increased with RelB depletion in hypoxia, however, this result is not significant (Appendix Fig. S6G). None of the selected cRel-dependent, hypoxia-downregulated genes were affected by cRel depletion in hypoxia, except *SOD1*, where cRel depletion gave the opposite effect to the RNA-seq analysis (Appendix Fig. S6H). Also, *LAMTOR2* was predicted to be both RelB- and cRel-dependent hypoxia regulated gene, however, RelB or cRel depletion showed no effect on its mRNA abundance (Appendix Fig. S6G,H). In conclusion, 5 out of 7 RelB-dependent, and 2 out of 2 RelA-dependent hypoxia downregulated genes were successfully validated by qPCR, however, none of the selected RNA-seq cRel-dependent genes had an effect on reversing the impact of hypoxia stimulation on the gene expressions. This indicates that RelA and RelB have a more dominant role in hypoxia-induced down-regulation of genes, compared to cRel. Interestingly, when we investigated the protein levels of DYRK1B and Vimentin by western blot (Appendix Fig. S7A–C), we confirmed that all NF-κBs tested were required for hypoxia-induced Vimentin, while cRel and, to a lesser extent, RelA were required for DYRK1B induction by hypoxia (Appendix Fig. S7A–C).

## NF-κB controlled hypoxia gene signatures in other cell types

Given that we had uncovered a variety of genes that were dependent on NF-κB for the response to hypoxia in HeLa cells,

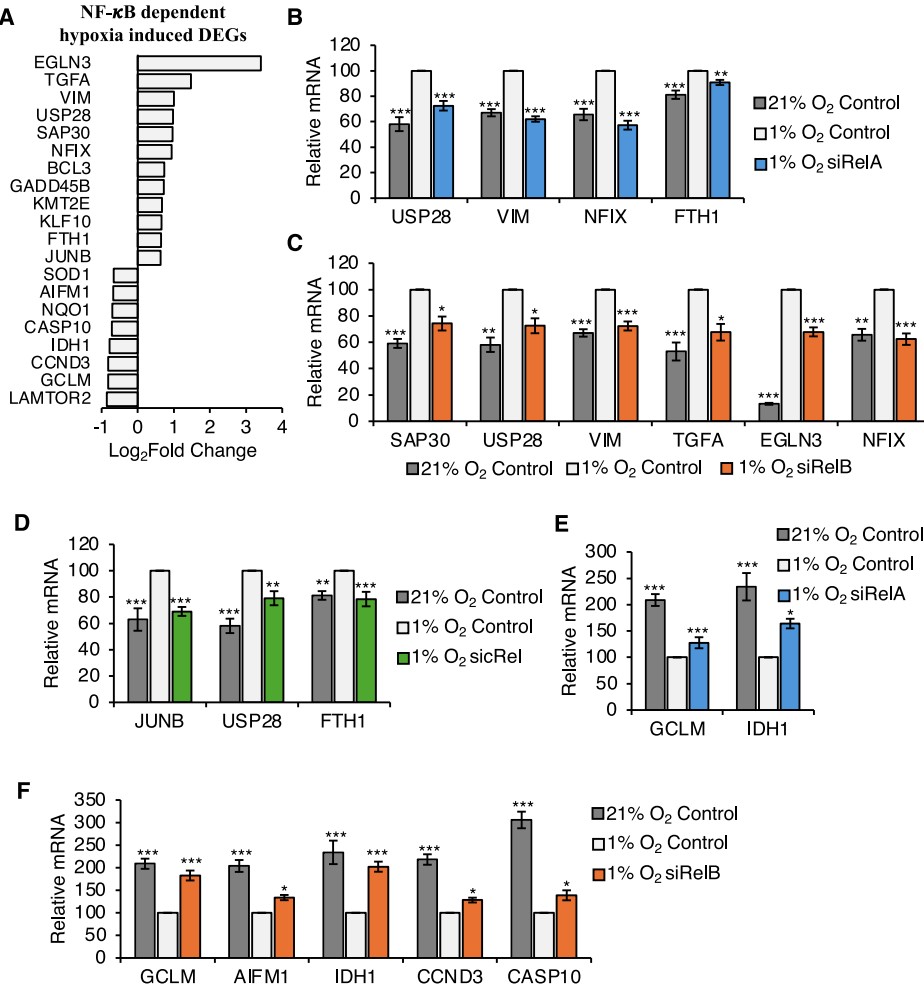

**Figure 3. Validation of NF-κB-dependent hypoxia inducible gene expression changes in HeLa cells.**

(A) Log₂Fold change of selected NF-κB dependent hypoxia up- and down-regulated DEGs showing altered transcript levels in hypoxia compared to normoxia identified by RNA-seq ($n = 2$). (B–F) qPCR analysis in HeLa cells cultured at 21% oxygen (normoxia) or 24 h 1% oxygen (hypoxia), transfected with control siRNA or RelA, RelB or cRel siRNAs. RelA- (B), RelB- (C), and cRel-dependent (D) hypoxia upregulated DEGs. RelA- (E) and RelB-dependent (F) hypoxia downregulated DEGs. Relative mRNA expression levels of the indicated genes were analysed using 18S as a normalising gene. Graphs show mean ($n \geq 3$) ± SEM, *$P < 0.05$, **$P < 0.01$, ***$P < 0.001$. Statistical significance was determined via one-way ANOVA with post-hoc Dunnett's test. Source data are available online for this figure.

we wanted to examine if these genes were also hypoxia-regulated in other cell types. To address this question, we analysed hypoxia (24 h, 1% oxygen) RNA-sequencing data in human cancer cell lines across different tissues, A549 (lung carcinoma), U87 (glioblastoma), HCT-116 (colorectal carcinoma), MCF-7 (breast adenocarcinoma), 501-mel (melanoma metastasis) and SKNAS (neuroblastoma) (Fig. 4A; Appendix Tables S2 and 3; Dataset EV3). Following quality control and the analysis pipeline (Appendix Fig. S8), volcano plots were created demonstrating the differentially expressed genes (DEGs) that were significantly changed (both upregulated or downregulated) across each cell line in response to hypoxia (Fig. 4B). Comparing these hypoxia gene signatures across different cell backgrounds with our HeLa-derived, NF-κB-dependent hypoxia signature revealed a very high overlap in cells such as U87, HCT116 and MCF-7 and more modest in SKNAS cells, across all NF-κB subunits (Fig. 4C; Appendix Fig. S9A). We also confirmed that a subset of these genes had been previously

described as direct NF-κB target genes in the Gilmore database (Fig. 4D; Appendix Fig. S9B). Finally, we were able to validate by qPCR analysis the vast majority of the NF-κB-dependent control over selected genes in U87, MCF-7 and A549 cells (Fig. 5A–L; Appendix Figs. S10A–F, S11A–C and S12A–Q; Appendix Table S1). Once again, RelB seemed to have the most widespread impact on hypoxia-dependent gene changes across all the cell systems investigated (Fig. 5A–L). Furthermore, some were also dependent on RelA, and very few were controlled by cRel (Fig. 5A–L). Taken together, these data indicate that the role of NF-κB in controlling hypoxia induced gene expression is common across a variety of cell lines.

To test if the NF-κB-dependent gene regulation that we observed in hypoxia, was also present under normal oxygen levels (normoxia), we performed qPCR experiments in HeLa cells with siRNA depletion of the NF-κB subunits followed by exposure to hypoxia or maintenance at normoxia (Appendix Fig. S13;

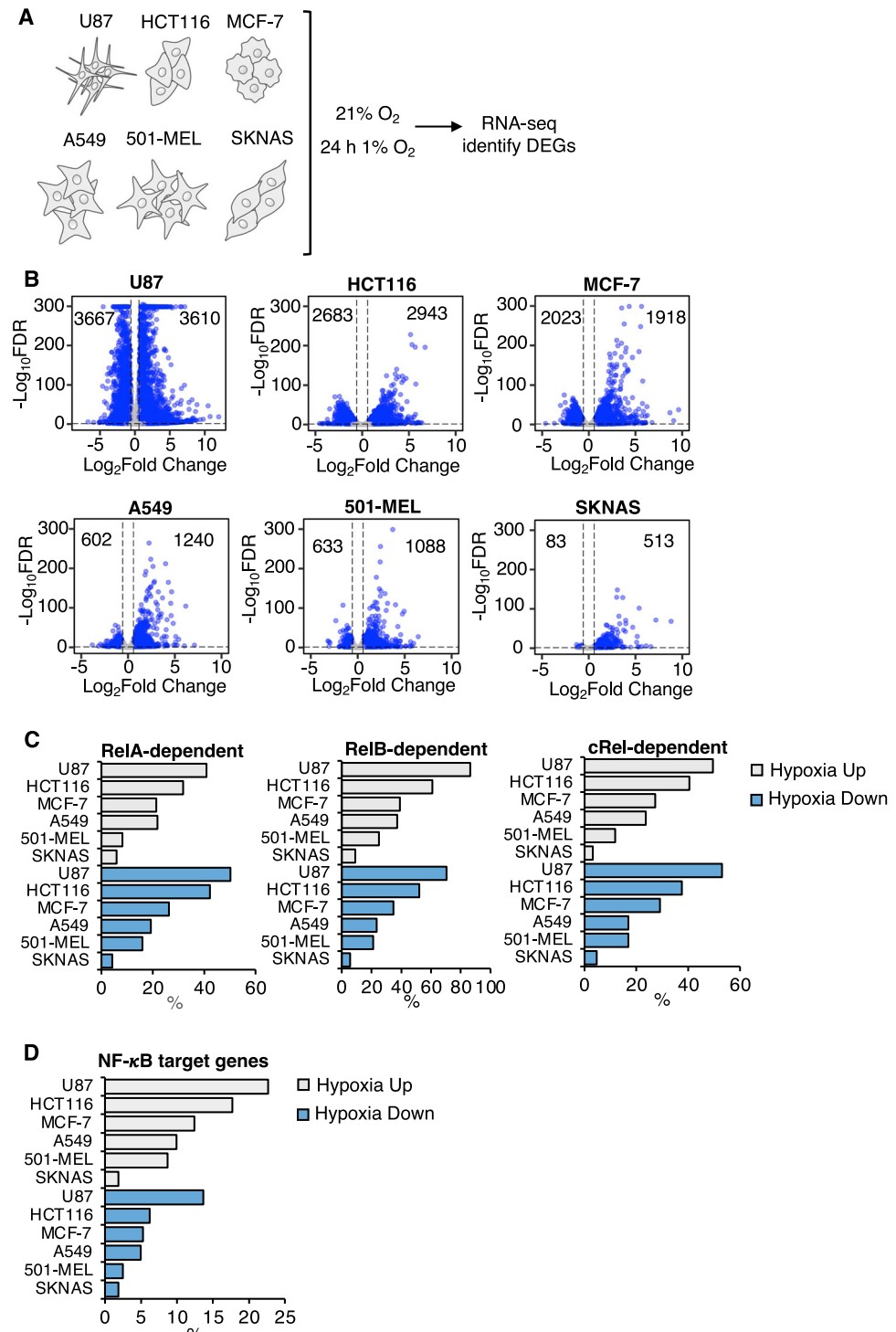

**Figure 4. Identification of NF-κB-dependent hypoxia inducible gene signature in different cell types.**

(A) RNA-seq ($n = 3$) in the indicated cell types cultured at 21% oxygen (normoxia) or exposed to 24 h 1% oxygen (hypoxia). (B) Volcano plots displaying differential expression analysis comparing 24 h hypoxia to control from RNA-seq datasets in the indicated cell lines. (C) Overlap of hypoxia inducible DEGs identified in different cell lines' RNA-seq datasets with NF-κB dependent hypoxia inducible DEGs identified in HeLa RNA-seq experiment following siRNA depletion of individual NF-κB subunits exposed or not to 24 h hypoxia. Percentage of NF-κB-dependent hypoxia inducible DEGs are displayed. (D) Overlap of hypoxia inducible DEGs identified in different cell lines' RNA-seq datasets with the Gilmore laboratory's NF-κB target genes. Percentage of NF-κB target genes are displayed. Source data are available online for this figure.

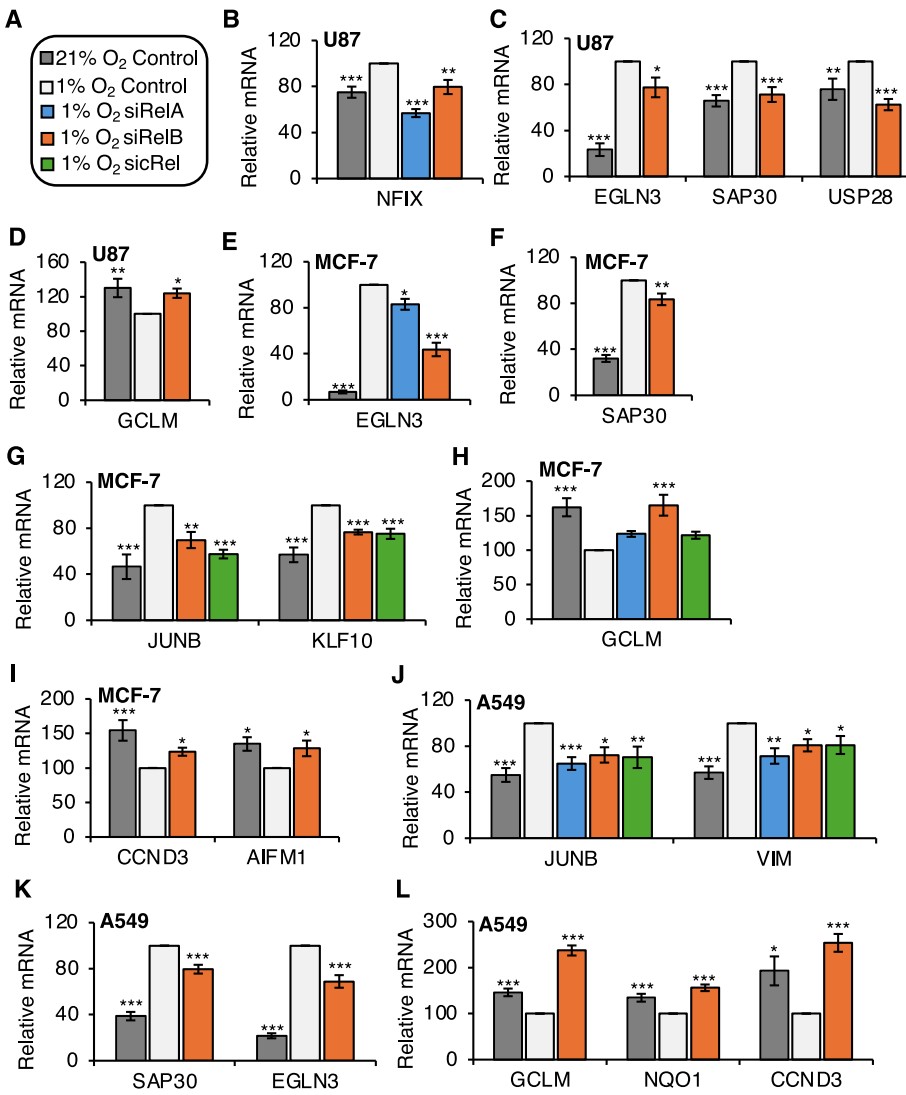

**Figure 5. Analysis of hypoxia inducible NF-κB-dependent genes across various cell backgrounds.**

(A) Key for qPCR analysis graphs. (B, C) Hypoxia upregulated NF-κB-dependent genes in U87 cells. (D) Hypoxia downregulated NF-κB-dependent genes in U87 cells. (E-G) Hypoxia upregulated NF-κB-dependent genes in MCF-7 cells. (H, I) Hypoxia downregulated NF-κB-dependent genes in MCF-7 cells. (J, K) Hypoxia upregulated NF-κB-dependent genes in A549 cells. (L) Hypoxia downregulated NF-κB-dependent genes in A549 cells. Relative mRNA expression levels of the indicated genes were analysed using Actin as a normalising gene for U87 and MCF-7 cell lines (B–I), and 18S for A549 cells. (J–L) Graphs show mean ($n \geq 3$) ± SEM, *$P < 0.05$, **$P < 0.01$, ***$P < 0.001$. Statistical significance was determined via one-way ANOVA with post-hoc Dunnett's test. Source data are available online for this figure.

Appendix Fig. S14). Firstly, we confirmed efficient depletion of *RelA*, *RelB* and *cRel* (Appendix Fig. S13A–C), and successful hypoxia treatment (Appendix Fig. S13D). A panel of six validated HeLa NF-κB-dependent hypoxia responsive genes were analysed, *VIM*, *USP28*, *EGLN3*, *SAP30*, *TFGA*, and *JUNB* (Appendix Fig. S14A–C). Each of these genes displayed NF-κB-dependent control of RNA levels in hypoxia, but not in normoxia (Appendix Fig. S14A–C). NF-κB-dependent gene regulation in normoxia and hypoxia was also accessed in A549 cells (Appendix Figs. S15 and S16). *RelA*, *RelB* and *cRel* were successfully depleted with siRNA treatments (Appendix Fig. S15A–C), and *CA9* levels were increased with hypoxia (Appendix Fig. S15D). As with HeLa cells, expression of each of the five validated A549 NF-

κB-dependent hypoxia responsive genes (*VIM*, *JUNB*, *EGLN3*, *SAP30*, and *GCLM*) were specifically regulated by NF-κB subunits in hypoxia (Appendix Fig. S16A–C). These data indicate, at least in the specific genes tested, that NF-κB-dependent regulation of hypoxia responsive genes, occurs under hypoxia but not basal oxygen levels.

As a complementary approach to siRNA depletion of NF-κB subunits, we also analysed NF-κB dependence of hypoxia responsive genes using qPCR in wild type and IKKα/β double knockout HCT116 CRISPR cell lines (Prescott et al, 2022) exposed to normoxia and hypoxia (Appendix Fig. S17). As positive control for hypoxia exposure, *CA9* RNA levels were increased with hypoxia (Appendix Fig. S17A). A panel of five genes (*EGLN3*, *JUNB*, *SAP30*,

*TGFA*, *USP28*, and *BCL3*) we identified and validated as NF-κB-dependent hypoxia responsive genes in HeLa cells, which were also hypoxia responsive in HCT116 cells (Appendix Fig. S10D), were analysed (Appendix Fig. S17B). Four of the six genes tested, displayed IKKα/β dependence for hypoxia induction in the HCT116 CRISPR cell lines, and IKKα/β-dependent control of gene expression was specific to hypoxia (Appendix Fig. S17B). These results, in which NF-κB was inhibited at the level of the IKKs, support our siRNA depletion analysis in a subset of identified NF-κB-dependent, hypoxia responsive genes, ruling out off target effects of the siRNA approach.

Given that all our analysis had been done in cancer cells, it is possible that role of NF-κB in hypoxia we have uncovered was restricted to cancer cells. To begin to address this aspect, we analysed Human umbilical vein endothelial cells (HUVECs) a non-transformed cell line for which RNA-sequencing data was available in the NCBI Geo database (Tiana et al, 2018). Although, the time of hypoxia exposure was less than the one we had previously used (16 h versus 24 h), these cells still mounted a robust hypoxia response at the transcriptional level (743 genes upregulated and 668 genes downregulated, Appendix Fig. S18A). When compared to our HeLa cell data, HUVECs shared 232 upregulated genes but only 82 downregulated genes (Appendix Fig. S18B,C). Comparison of the NF-κB-dependent hypoxia-inducible gene from HeLa and HUVEC cells indicates that NF-κB is required for 45% of hypoxia repressed genes HUVECs share with HeLa cells, and 28% of hypoxia induced genes (Appendix Fig. S18D,E).

HUVECs have also previously been analysed for their response to TNF-α, a canonical NF-κB activating stimulus (Fowler et al, 2022). Furthermore, this study also determined the role of RelA in HUVEC responses to TNF-α (Fowler et al, 2022). Using these datasets, we made a series of observations. HUVECs are highly responsive, at a transcriptional level, to TNF-α, with 1640 genes upregulated after 4 h and 1445 genes upregulated after 10 h, 1078 of which were already upregulated after 4 h (Appendix Fig. S19A,C, Dataset EV4). Similarly, 1687 genes were downregulated after 4 h TNF-α and 1722 genes were downregulated after 10 h TNF-α, with 990 genes shared between the treatment periods (Appendix Fig. S19B,D, Dataset EV4). Depletion of RelA in this cell system revealed that it was needed for 40% of genes upregulated at 4 h and 49.6% of genes upregulated after 10 h TNF-α (Appendix Fig. S19E). Similarly, RelA was needed for 43.2% of TNF-α downregulated genes after 4 h and 40.5% of downregulated genes after 10 h (Appendix Fig. S19F). This indicates that RelA has a significant role in the response of HUVEC cells to TNF-α, controlling gene activation and repression. We next compared the RelA-dependent TNF-α regulated genes to our RelA-dependent hypoxia gene signature obtained from HeLa cells. This revealed that 14 of the RelA-dependent TNF-α-induced genes were also observed to be RelA-dependent in hypoxia (Appendix Fig. S19G). When we analysed the downregulated genes, 15 genes were dependent on RelA for their repression in both Hypoxia and TNF-α (Appendix Fig. S19H). Interestingly, 28 genes that require RelA for repression in TNF-α, require RelA for their induction in hypoxia, while 6 genes that required RelA for TNF-α induction required RelA for hypoxia repression. Taken together this data supports our model, while for some genes, hypoxia works similarly to a canonical inducer of NF-κB, more often hypoxia utilises NF-κB to achieve the opposite result in the control of gene expression, even in non-cancer cell types.

## NF-κB is important for hypoxia induced cellular responses

Given the extent of the NF-κB contribution to the hypoxia transcriptional response (35% of activated and 62% of repressed genes), we next determined if any of the cellular responses to hypoxia were altered in the absence of NF-κB subunits. Pathway analysis of the genes repressed in hypoxia that depend on NF-κB indicated Reactive Oxygen Species (ROS) and Oxidative phosphorylation as two of the main hallmarks altered. To validate this at the cellular level, we measured ROS production using a molecular dye (CellRox); we have previously used this to demonstrate that hypoxia induces ROS generation in these cells (Batie et al, 2019) as described in other systems (Alva et al, 2024). As expected, hypoxia exposure resulted in a significant increase in ROS (Fig. 6A,B). Importantly, depletion of RelA, and to a lesser extent RelB and cRel, reduced the ROS levels generated by hypoxia (Fig. 6A,B). As this ROS probe primarily detects superoxide in cells (McBee et al, 2017), we investigated genes specifically associated with detoxifying superoxide in cells. One such gene is superoxide dismutase 1, *SOD1*. Although, we could not demonstrate a statistically significant change in levels of mRNA for *SOD1* when NF-κB subunits were depleted in hypoxia (Appendix Fig. S6G,H), Western blot analysis confirmed that hypoxia does lead to reduced SOD1 expression (Fig. 6C; Appendix Fig. S20), and that RelB, and to lesser extent cRel, relieved this repression when depleted (Fig. 6C,D). These results suggest that ROS generation in hypoxia requires NF-κB via a mechanism that partially requires SOD1 reduction. We also investigated if NRF2 is induced in hypoxia as marker of oxidative stress (Appendix Fig. S21). NRF2 levels were unchanged in response to 1 and 24 h of 1% oxygen exposure (Appendix Fig. S21A) and only a small number of NRF2 target genes (Hayes and Dinkova-Kostova, 2014; Morgenstern et al, 2024) were upregulated in hypoxia from our RNA-seq analysis (Appendix Fig. S21B), thus there is limited evidence for induction of NRF2, at least at the time and level of hypoxia exposure employed here, suggesting the level of ROS induced by hypoxia may be important for signalling but may not activate the NRF2 stress response.

Similarly, we investigated a panel of proteins involved in oxidative phosphorylation (Fig. 7, Appendix Fig. S22). This panel contained antibodies for proteins such as ATP Synthase F1 subunit alpha (ATP5A); Ubiquinol-cytochrome C reductase core protein 2 (UQCRC2), Succinate Dehydrogenase subunit B (SDHB) and Isocitrate Dehydrogenase 1 (IDH1). Hypoxia led to reductions in several proteins represented in this panel, such as ATP5A, UQCRC2, NDUFB8, COX2 and IDH1 (Fig. 7A,B; Appendix Fig. S22A–E). Depletion of NF-κB led to reversal of the protein levels to those of normoxia or slightly above for IDH1 (Fig. 7A,B; Appendix Fig. S22A–E). These data support our RNA-seq analysis, where oxidative phosphorylation was a hallmark that was significantly repressed in hypoxia, dependent on NF-κB presence (Fig. 2G). To confirm that hypoxia-dependent repression of protein levels of oxidative phosphorylation pathway components, we also examined HCT116 and their IKKα/IKKβ CRISPR knockout derivatives (Prescott et al, 2022) (Fig. 7C). In these cells, where NF-κB signalling is abolished, hypoxia exposure did not result in reduced levels of ATP5A, UQCRC2, COX2 and NDUFB8 (Fig. 7C). Only one of the IKKα/IKKβ KO clones reversed SOD1 repression in hypoxia (Fig. 7C).

NF-κB was previously shown to actively repress gene expression via HDAC activity (Campbell et al, 2004; Elsharkawy et al, 2010; Rocha et al, 2003). To test if some of NF-κB-repressed genes also require

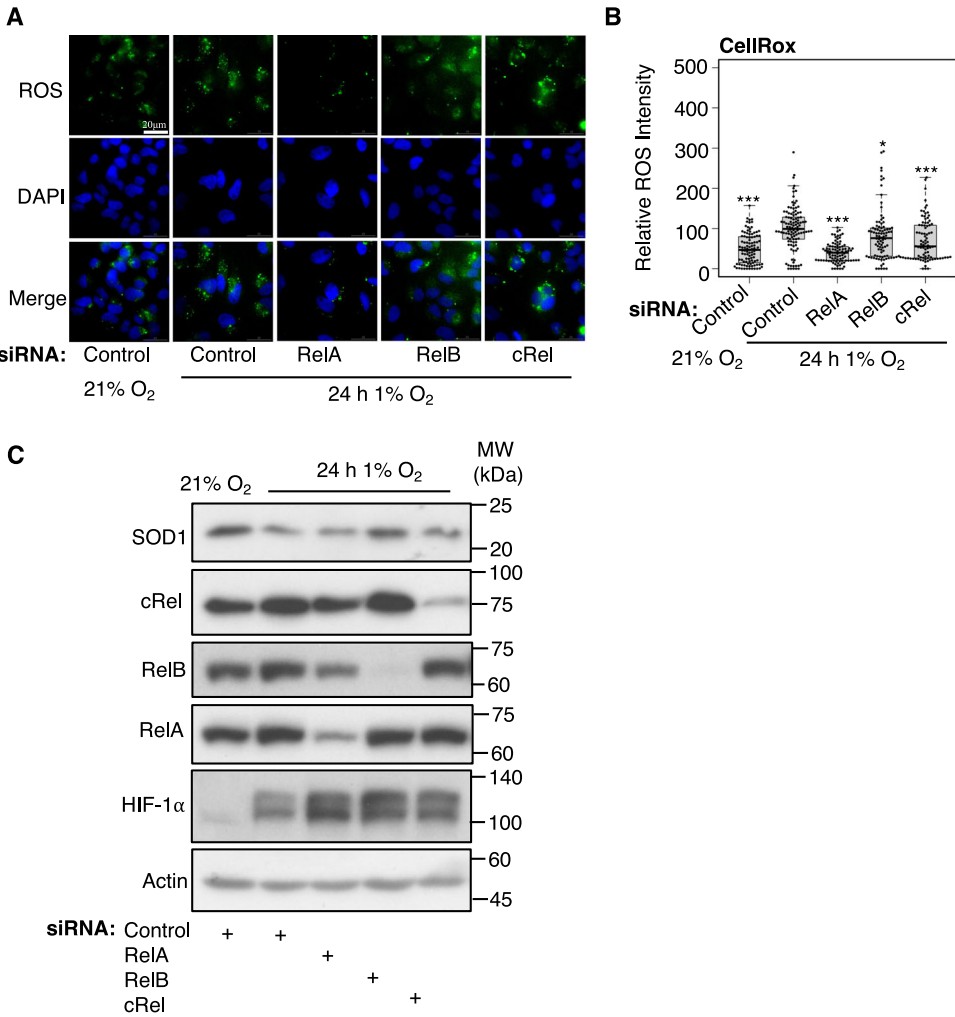

**Figure 6. NF-κB is required for ROS generation in hypoxia.**

(A) Cellular ROS measurements using CellRox staining and immunofluorescence analysis in HeLa cells, cultured at 21% oxygen (control) or 24 h 1% oxygen (hypoxia), transfected with control siRNA or RelA, RelB or cRel siRNAs. Representative images from 3 independent experiments are shown. (B) Quantification of relative ROS signal in individual cells from 3 biological replicates, represented as a beeswarm with box plot, *P < 0.05, **P < 0.01, ***P < 0.001 (compared to hypoxia control siRNA treated cells). Statistical significance was determined via one-way ANOVA with post-hoc Dunnett's test. (C) Immunoblot analysis of the indicated protein in HeLa cells exposed or not to 1% oxygen for 24 h, with siRNA transfection of control, RelA, RelB or cRel. Source data are available online for this figure.

HDAC activity, we treated HeLa cells with an HDAC inhibitor, Saha, alone or in combination with hypoxia. This analysis revealed that while HDAC inhibition reversed hypoxia induced repression of ATP5A, UQCRC2 and COX2 (Fig. 7D), it had no effect on hypoxia-induced repression of SOD1 (Appendix Fig. S23), causing SOD1 levels to reduce in normoxia. Taken together, these results suggest that NF-κB represses gene expression in hypoxia by mechanisms requiring HDAC activity and additional unknown repressors.

## Discussion

Here, we have investigated the contribution NF-κB makes to the transcriptional response to hypoxia. We find that NF-κB is required for induction of genes in hypoxia but seems especially important for gene repression under these conditions (Fig. 8).

NF-κB pathway activation in hypoxia has been well documented across cells, tissues and organisms (Bandarra et al, 2014; Chandel et al, 2000; Cummins and Taylor, 2005; D'Ignazio and Rocha, 2016; Koong et al, 1994; Taylor et al, 1999; Van Welden et al, 2017). While the exact mechanism of activation may depend on the system and timing of the investigation, it was implied that NF-κB is part of the cellular response to hypoxia. However, to what extent NF-κB contributes to the transcriptional response to hypoxia was not known. Our understanding of hypoxia is dominated by the HIFs, whose role as a master regulator of the transcriptional response is well established from direct chromatin binding (Batie et al, 2022), to a conserved list of genes consistently induced in hypoxia (Puente-Santamaria et al, 2022). Our analysis of depletion of individual Rel subunits reveals that NF-κB is required for 35% of all hypoxia-induced genes but strikingly is required for over 60% of all hypoxia-repressed genes. Interestingly, RelB seems as important

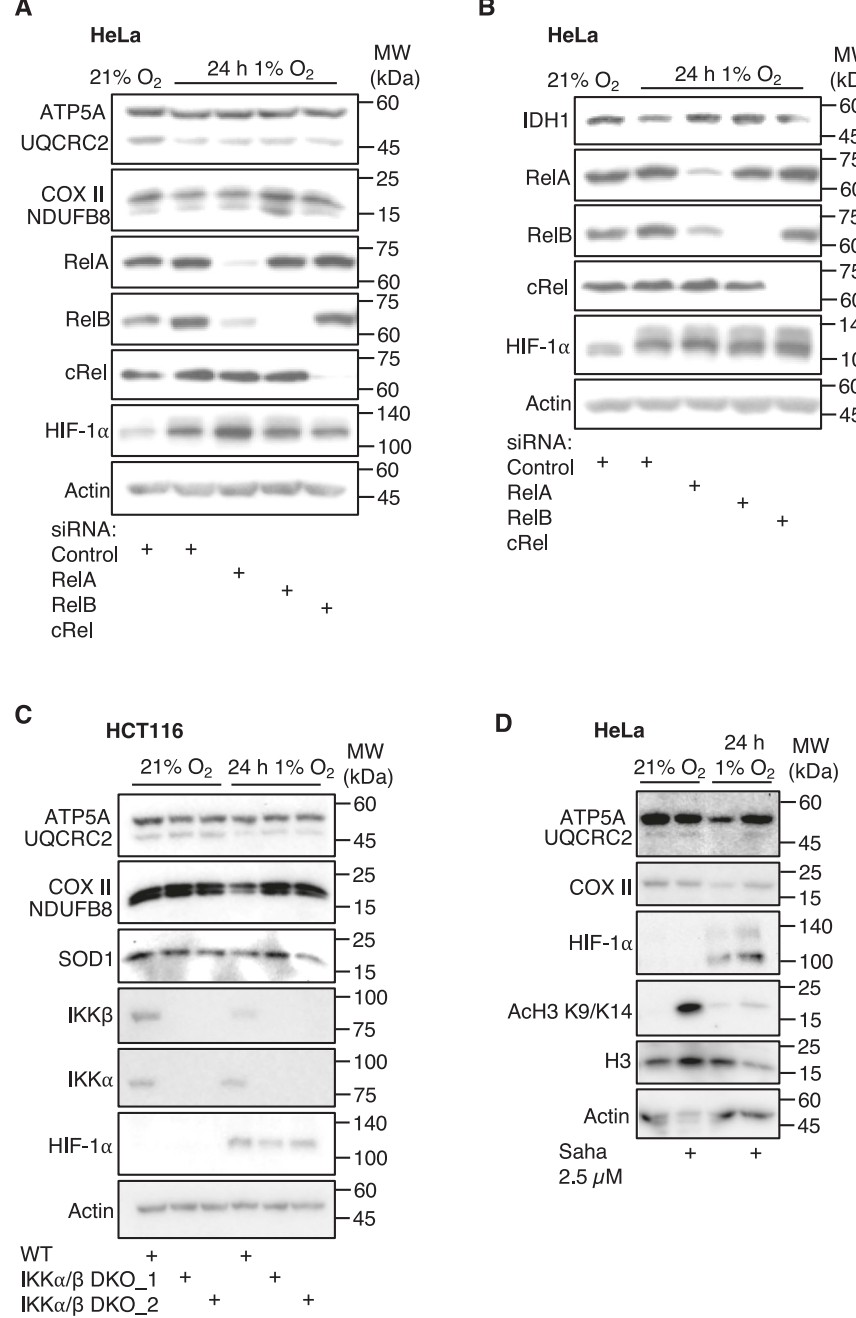

**Figure 7. NF-κB is required for expression of oxidative phosphorylation-related proteins in hypoxia.**

(A, B) Immunoblot analysis of the indicated proteins in HeLa cells cultured at 21% oxygen (control) or 24 h 1% oxygen (hypoxia), transfected with control siRNA or RelA, RelB or cRel siRNAs. Representative images from 3 independent experiments are shown. (C) HCT-116 wild-type (WT) and IKKα/IKKβ double knockout cells, were cultured at 21% oxygen (control) or 24 h 1% oxygen (hypoxia) prior to cell lysis. Western blot was performed with the indicated antibodies. (D) Immunoblot analysis of the indicated proteins in HeLa cells cultured at 21% oxygen (control) or 24 h 1% oxygen (hypoxia), with or without 2.5 μM Saha (HDAC inhibitor) for 24 h. Representative images from 3 independent experiments are shown. Source data are available online for this figure.

as RelA, something previously not appreciated in the response to hypoxia. Very few studies have investigated RelB's role in hypoxia (Oliver et al, 2009; Patel et al, 2017; Riedl et al, 2021). These studies have mostly focused on either gene induction (Patel et al, 2017), short period of hypoxia (Oliver et al, 2009) or used a hypoxia mimetic as inducer (Riedl et al, 2021). None of these studies

investigated RelB-mediated gene regulation in hypoxia, making our study one of the first to use an unbiased approach to analyse the contribution of RelB to the transcriptional profile of cells exposed to hypoxia. Of note, RelA is able to directly regulate the expression of other NF-κB subunits directly (Oeckinghaus and Ghosh, 2009), so the effects of RelA may represent a contributions of other NF-κB

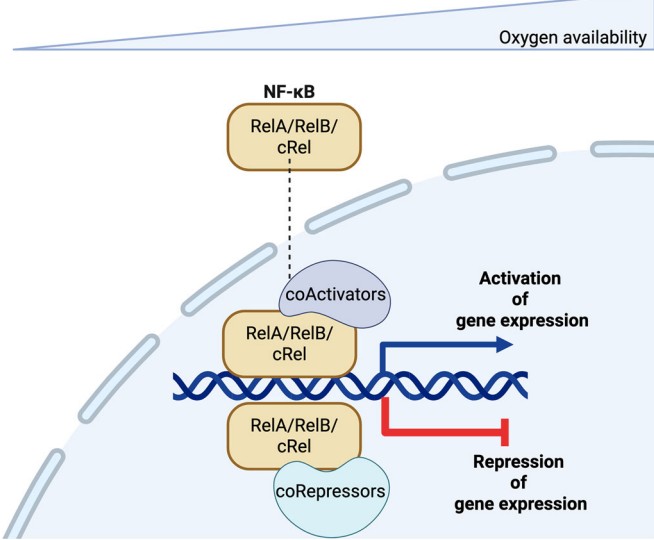

**Figure 8. NF-κB is a crucial transcription factor in hypoxia.**

Upon exposure to hypoxia, or reduced oxygen availability, NF-κB associates with co-activators to induce gene expression. However, NF-κB can also associate with co-repressors, leading to significant repression of gene expression under hypoxic conditions.

subunits as well. Interestingly, NF-κB-dependent hypoxia signatures are more widely regulated across cell types than validated NF-κB targets, suggesting that perhaps NF-κB-dependent genes in hypoxia are different from standard cytokine induced ones. Although analysis of the ChIP-Atlas database, confirmed that 80% of the genes we have identified to be NF-κB-dependent have NF-κB subunits bound to them, a similar NF-κB binding percentage was present in genes found to be independent of NF-κB. Further analysis performing ChIP-seq or CUT&RUN in hypoxia for NF-κB subunits would help distinguish between direct and indirect regulation of genes by NF-κB in hypoxia. It is also important to acknowledge that NF-κB is critical to maintain HIF gene expression in different cells and organisms suggesting a wider role for NF-κB in underpinning HIF activity (D'Ignazio and Rocha, 2016; Kenneth et al, 2009; Rius et al, 2008; van Uden et al, 2008; van Uden et al, 2011). Interestingly, both transcription factors are evolutionarily conserved and present in the primitive organism *Trichoplax adhaerens* (Loenarz et al, 2011; Romanova and Moroz, 2024; Williams and Gilmore, 2020). Although their functional crosstalk has not been investigated in *T. adhaerens*, it is conserved in *Drosophila melanogaster* (Bandarra et al, 2014; Bandarra et al, 2015; van Uden et al, 2011), indicating an ancient association between these two pathways.

In our analysis in HeLa cells, individual depletion of NF-κB subunits only resulted in around 10–20% reduction in HIF-1α mRNA (Appendix Fig. S24) and no visible changes to HIF-1α protein levels after 24 h hypoxia exposure. This supports our original findings that more than one NF-κB subunit is required to control HIF-1α levels (van Uden et al, 2008). Furthermore, it is possible that NF-κBs control over HIF-1α is observed at earlier response times of hypoxia, as after 24 h of hypoxia HIF-1α protein levels are already reduced in most cell types. Nevertheless, for genes

known to be HIF-dependent, NF-κB's function could be indirect. This is also supported by the analysis of HIF ChIP-seq data where there was a positive correlation between HIF binding and upregulation of genes that are also NF-κB dependent. However, a more thorough investigation of the interplay between HIF and NF-κB, across a range of timepoints in response to hypoxia, is required.

Unlike HIF, NF-κB is able to actively repress genes (Burkhart et al, 2005; Campbell et al, 2004; Elsharkawy et al, 2010; Grigoriadis et al, 1996; Saha et al, 2008) by actively recruiting repressor proteins such as HDACs (Elsharkawy et al, 2010), or by engaging with repressor complexes such as EZH2 directly (Dardis et al, 2023) or via nuclear sumoylated-IκBα (Mulero et al, 2013). These effects have been observed in response to a variety of stimuli or in developmental stages of differentiation (Burkhart et al, 2005; Campbell et al, 2004; Elsharkawy et al, 2010; Grigoriadis et al, 1996; Mulero et al, 2013; Saha et al, 2008). However, the role NF-κB in gene repression is still under investigated, despite its importance when studied (Campbell et al, 2004; Grigoriadis et al, 1996; Mulero et al, 2013). Our results reveal the extent of NF-κB contribution to the hypoxia transcriptional response, highlights the importance of NF-κB during such a critical developmental and pathological stimulus. Our results suggest that NF-κB may repress gene expression via HDAC activity. However, HDAC inhibition did not reverse hypoxia induced repression for all the proteins we investigated. In the future, it will be important to determine the exact mechanism by which NF-κB mediates its repressive effects in hypoxia as our results indicate that HDACs are only a part of the mechanism.

The hallmark signatures for hypoxia repressed genes controlled by NF-κB were ROS signalling and Oxidative Phosphorylation. Although distinct, these signatures are intrinsically linked in hypoxia (Hamanaka and Chandel, 2010; Solaini et al, 2010). Given the reduction in oxygen, mitochondria are considered one of the sources of ROS due to alteration in the electron transport chain activity (Alva et al, 2024; Kracht et al, 2020; Solaini et al, 2010). Furthermore, reducing the expression of genes associated with oxidative phosphorylation in time of oxygen deficiency is a cost-effective mechanism to conserve energy (Kracht et al, 2020). Although technically we are unable to measure oxygen consumption in hypoxia, we could determine that several proteins associated with this function were reduced in hypoxia and that NF-κB depletion reversed this repression. This might be, in the long term, detrimental for the cell, as cell death could be a consequence of prolonged NF-κB depletion in hypoxia.

NF-κB has been previously shown to translocate to mitochondria as well (Ivanova and Perkins, 2019), a process important for ROS generation. Our data supports NF-κB's role in generation of ROS in response to hypoxia. ROS's role in hypoxia is multifaceted (Alva et al, 2024; Hamanaka and Chandel, 2010), and hence it contributes not only to pathology but is also used as a signalling molecule. However, these roles seem to be quite dependent on the extent of ROS generation cell type-specific (Alva et al, 2024).

Our study has several limitations to consider. For this study we only used cancer cells and given NF-κB's pleotropic effects, it is possible that in non-transformed cells, NF-κB has a different role. We analysed a non-transformed cell type, HUVECs, with available transcriptomics in hypoxia (Tiana et al, 2018) and a known NF-κB inducer, TNF-α (Fowler et al, 2022). This analysis also correlated well with our cancer cell analysis, revealing that NF-κB is important

for repression of genes in hypoxia but also for some of the hypoxia induced genes. Another limitation of our study is that we only assessed the role of NF-κB in hypoxia and not at basal oxygen conditions in our transcriptomic analysis. Our qPCR analysis demonstrated that NF-κB-dependent regulation of hypoxia responsive genes occurs under hypoxia but not basal oxygen levels in the specific genes tested. However, further RNA-sequencings experiment with depletion or inhibition of NF-κB subunits in normoxia and hypoxia is required to properly delineate hypoxia-specific effects. Similarly, analysis of HUVECs revealed that only a subset of TNF-α induced/repressed genes are also controlled by RelA under non-stimulated conditions. Another limitation is the use of only one siRNA sequence per NF-κB subunit. While were careful to validate and quantify NF-κB subunit knockdown in each case we also validated our results with additional orthogonal approaches to inhibit NF-κB subunit such as the IKKα/IKKβ CRISPR knockout cells. Nonetheless, it is still possible that some off target effects are present; however, this is the same with any experimental approach. Despite these limitations, our study uncovers a pivotal role for NF-κB in regulating the transcriptional response to hypoxia. Strikingly, we find that NF-κB's primary function under these conditions is to mediate transcriptional repression (Fig. 8), a contribution that has been largely underappreciated. While previous studies have highlighted the importance of this mechanism (Batie et al, 2018; Cavadas et al, 2017), our understanding of how gene repression is orchestrated in hypoxia remains limited. This work takes a critical step toward filling that gap.

## Methods

### Reagents and tools table

| Reagent/resource | Reference/source | Identifier/catalogue number |
| --- | --- | --- |
| **Cell lines** | | |
| HeLa | ATCC | CRM-CCL-2 |
| A549 | ATCC | CRM-CCL-185 |
| SKNAS | ATCC | CRL-2137 |
| U87 | Merck | 89081402 |
| MCF-7 | Merck | 86012803 |
| HCT116 IKKα/β double knockout (DKO) | (Prescott et al, 2022) | |
| **siRNAs (gene target)** | | **Sequence** |
| RelA | IDT | GCCCUAUCCCUUUACGUCA |
| RelB | IDT | GGAUUUGCCGAAUUAACAA |
| cRel | IDT | ACAGCUGAAUGAUAUUGAA |
| Control | IDT | CAGUCGCGUUUGCGACUGG |
| **Antibodies** | | |
| RelA | Santa Cruz | sc-8008 |
| RelB | Cell Signalling | 10544 |
| cRel | Cell Signalling | 4727 |
| HIF-1α | BD Biosciences | 610958 |
| β-Actin | Proteintech | 66009-1-Ig |
| IDH1 | Proteintech | 12332-1-AP |
| SOD1 | Santa Cruz | sc-17767 |

| Reagent/resource | Reference/source | Identifier/catalogue number |
| --- | --- | --- |
| Total Oxphos (ATP5A, UQCRC2, and SDHB) | Abcam | ab110411 |
| NRF2 | Cell Signalling | 12721 |
| BNIP3L | Cell Signalling | 12396 |
| Vimentin | Cell Signalling | 5741 |
| DYRK1B | Cell Signalling | 5672 |
| Anti-mouse IgG, HRP-Linked | Cell Signalling | 7076 |
| Anti-rabbit IgG, HRP-Linked | Cell Signalling | 7074 |
| **RT-qPCR primers** | **Forward** | **Reverse** |
| Actin | CTGGGAGTGGGTGGAGGC | TCAACTGGTCTCAAGTCAGTG |
| 18S | AAACGGCTACCACATCCAAG | CGCTCCCAAGATCCAACTAC |
| CA9 | CTTTGCCAGAGTTGACGAGG | CAGCAACTGCTCATAGGCAC |
| RelA | CTGCCGGGATGGCTTCTAT | CCGCTTCTTCACACACTGGAT |
| RelB | TCCCAACCAGGATGTCTAGC | AGCCATGTCCCTTTTCCTCT |
| cRel | CTGCCTTCTTCAAGCTGGTC | CGCTTCCATTCCGACTATGT |
| GADD45B | TCGGCCAAGTTGATGAATG | TGAGCGTGAAGTGGATTTG |
| JUNB | GACCAAGAGCGCATCAAA | TCTTCACCTTGTCCTCCA |
| NFIX | TGTTGATGACGTGTTCTATCC | CCAGCTTTCCTGACTTCTTTA |
| USP28 | GAAGTAGAGGAGTGGGAAGA | GAAGGCTCTTGTGATGTAGAG |
| KMT2E | GCCAACTGCCCTACATAAA | GCTCTTTACCCAGGAAGAATAC |
| TGFA | GCATGTGTCTGCCATTCT | GTGATGGCCTGCTTCTTC |
| VIM | CCAGCTAACCAACGACAAA | TCCTCTCTCTGAAGCATCTC |
| SAP30 | GAGCGCAAGGCATCTTTA | AATCACCTCCATCATCATCAC |
| EGLN3 | CTACGTCAAGGAGAGGTCTAA | CAGATAGTAGATGCAGGTGATG |
| KLF10 | TGCCTTCGTGTTGAAATCC | GGGCGCGATTATGCAATTA |
| BCL3 | CTCTCCATATTGCTGTGGTG | TGTCTGCCGTAGGTTGT |
| FTH1 | AGTGCCGTTGTTCAGTTC | AGACAGCCACACCTTAGT |
| GCLM | CTGTTCAGTCCTTGGAGTTG | CTCCCAGTAAGGCTGTAAATG |
| NQO1 | CCTGGAAGGATGGAAGAAAC | GAATCCTGCCTGGAAGTTTAG |
| AIFM1 | CCAGCCACCTTCTTTCTATG | CCATGTTGTCTCTCACATCC |
| IDH1 | GCAGTACAAGTCCCAGTTT | TGAAGCCTCCCTCTGATT |
| SOD1 | TCGAGCAGAAGGAAAGTAATG | CCTGCTGTATTATCTCCAAACT |
| CCND3 | GATTTCCTGGCCTTCATTCT | GGGTACATGGCAAAGGTATAA |
| CASP10 | CGTATCAAGGAGAGGAAGAAC | TGTGGCTCTGTTACCATTAC |
| LAMTOR2 | CACCCTGCTGCTGAATAA | CCCAGATGTTACTGGCTATG |
| **Chemicals, enzymes, and other reagents** | | |
| DMEM | Thermo Fisher Scientific | 41966029 |
| FBS | Merck | F7524 |
| L-glutamine | Thermo Fisher Scientific | 25030024 |
| Penicillin + Streptomycin | Thermo Fisher Scientific | 15140122 |
| Mycoplasma Detection Kit | Lonza | LT07-318 |
| Bardoxolone methyl (CDDO-Me) | Merck | TA9H9A9A743D |
| SAHA (HDAC inhibitor) | Merck | SML0061 |
| Interferin | Polyplus | 101000016 |
| RNeasy mini kit | Qiagen | 74104 |
| iscript reverse transcription kit | Bio-Rad | 1708891 |

| Reagent/<br>resource | Reference/source | Identifier/catalogue number |
|---|---|---|
| SYBR green | Thermo Fisher Scientific | 4364344 |
| CellROX green | Thermo Fisher Scientific | C10444 |
| **Software** | | |
| STAR v2.7.11b | Dobin et al, 2013 | |
| Subread v2.0.1 | Liao et al, 2013 | |
| RStudio v4.3.3 | Posit | |
| HOMER v5.1 | Heinz et al, 2010 | |
| WebGestalt | Liao et al, 2019 | |
| Flaski/ venn<br>diagram | Iqbal, 2021 | |
| BioRender | https://BioRender.com/ | |
| ImageJ v1.53a | NIH | |
| SPSS v30 | IBM SPSS statistics | |
| OMERO v5.7.2 | Allan et al, 2012 | |

## Methods and protocols

### Cell culture

Human cervical carcinoma HeLa, lung carcinoma A549, brain glioblastoma U87, breast adenocarcinoma MCF-7, and IKKα/β double knockout (DKO) HCT116 cell lines were maintained at 5% $CO_2$ and 37 °C in Dulbecco's modified Eagle's medium (DMEM) supplemented with 10% v/v foetal bovine serum (FBS), 1% L-glutamine, and 1% penicillin-streptomycin. Cell lines were cultured not more than 30 passages and routinely tested for mycoplasma contamination using MycoAlert Mycoplasma Detection Kit.

### Treatments and siRNA transfections

Hypoxia treatments were performed by incubating cells in an InVivo300 Hypoxia Workstation (Baker Ruskin, UK) at 1% $O_2$, 5% $CO_2$ and 37 °C. For stabilisation of NRF2, cells were treated with 100 nM of Bardoxolone methyl (CDDO-Me) for 16 h, and 2.5 μM of Saha (HDAC inhibitor) for 24 h. Cells were transfected with 27 nM of small interfering RNA (siRNA) oligonucleotides for 48 h using Interferin transfection reagent according to manufacturer's instructions.

### RNA extraction and RNA-seq data analysis

RNA was extracted using the RNeasy Mini Kit following manufacturer's instructions. RNA-seq was performed on a Novaseq 6000 (Illumina, UK) with paired-end 150 bp run type. Sequencing reads were aligned to the human genome assembly version hg38 (GRCh38, Ensemble) using STAR sequence aligner (Dobin et al, 2013) to generate coordinate-sorted binary alignment map (bam) files. Read counts for each transcript (GRCh38, Ensembl) were generated using the featureCounts function of Subread (Liao et al, 2013). Differential expression analysis was performed using R Bioconductor package DESeq2 (Love et al, 2014) with filtering for effect size (>±0.58 log₂ fold change) and statistical significance (FDR < 0.05) to determine differentially expressed genes (DEGs). Sample to sample distance comparison heatmaps using rlog transformed read count data were generated using R Bioconductor packages DESeq2 and pheatmap. PCA plot of rlog transformed read count data were generated using R Bioconductor packages

DESeq2 and ggplot2. Volcano plots for display of differential expression analysis were generated using R Bioconductor package EnhancedVolcano. Z-score heatmaps of VST transformed read count data were generated using DESeq2 and pheatmap. RelA, RelB, and cRel-dependent and -independent hypoxia responsive genes were determined by overlapping hypoxia control siRNA DEGs (compared to normoxia control siRNA) with hypoxia RelA, RelB, and cRel siRNA DEGs (compared to normoxia control siRNA). NF-κB-dependent and -independent hypoxia responsive genes were determined by overlapping hypoxia control siRNA DEGs (compared to normoxia control siRNA) with the combined list of RelA, RelB, and cRel siRNA DEGs (compared to normoxia control siRNA). Overrepresentation analysis (ORA) was performed using WEB-based Gene SeT AnaLysis Toolkit (WebGestalt) (Liao et al, 2019) in overrepresentation analysis (ORA) mode using the Molecular Signatures Database (MSigDB) hallmark gene sets (Liberzon et al, 2015; Subramanian et al, 2005). Motif enrichment analysis was performed using the findMotifs tool of HOMER (Heinz et al, 2010) with gene promoters defined as 400 bps upstream to 100 bps downstream of transcription start sites.

### qPCR

RNA was extracted using the RNeasy Mini Kit following manufacturer's instructions. RNA was converted to cDNA using iscript reverse transcription kit and qPCR analysis of transcript expression changes was performed by running 3 μl of DNA on a QuantStudio 1 qPCR platform (Applied Biosystems) with power-track Sybr green reaction mix in a final reaction of 15 μl. The quantity of mRNA was determined using ΔΔCT method and normalised by Actin or 18S used as reference genes.

### Statistical analysis

For qPCR analysis comparing two conditions, statistical significance was determined via Student's t-test. For qPCR analysis comparing more than two conditions, statistical significance was determined via one-way ANOVA with post-hoc Dunnett's test or Tukey's HSD test. For overlap of differentially expressed genes (DEGs) identified by RNA-seq, statistical significance was determined via hypergeometric test using Venn diagram function of Flaski online data analysis tool with its default parameters (Iqbal, 2021). For all other statistical analysis, default settings of the analysis tool were used. In all cases, $*P < 0.05$, $**P < 0.01$, $***P < 0.001$.

### Data mining of publicly available datasets

HeLa HIF ChIP-seq dataset (Ortmann et al, 2021) (GSE169040) was downloaded from the Gene Expression Omnibus (Edgar et al, 2002). MCF-7 (GSE153291), A549 (GSE186370), HCT116 (GSE81513), U87 (GSE78025), and 501-MEL (GSE132624) hypoxia and HUVEC TNF-α (Fowler et al, 2022) (GSE201466) RNA-seq datasets were downloaded from the Gene Expression Omnibus (Edgar et al, 2002) and processed using the RNA-seq data analysis pipeline described above. HUVEC hypoxia RNA-seq data was obtained from (Tiana et al, 2018). NRF2 target genes were obtained from Hayes and Dinkova-Kostova (2014) and Morgenstern et al (2024). NF-κB target genes were downloaded from the Gilmore laboratory website and converted to Ensembl gene annotation format using R Bioconductor package biomaRt (Durinck et al, 2005).

## Cellular reactive oxygen species detection

Cells were fixed and cellular reactive oxygen species (ROS) were measured using CellROX Green reagent according to the manufacturer's instructions. Cells were imaged using an Epifluorescence microscopy (Zeiss Axio Observer Z.1). Images were quantified using OMERO open microscopy environment (Allan et al, 2012). Scale bars = 20 μm.

## Immunoblots

Cells were lysed in Radio Immunoprecipitation Assay (RIPA) lysis buffer (50 mM Tris-HCl pH 8.0, 150 mM NaCl, 1% v/v NP40, 0.25% w/v Na-deoxycholate, 0.1% w/v SDS, 10 mM NaF, 2 mM $Na_3VO_4$, and 1 protease inhibitor tablet (Thermo Fisher Scientific) per 10 mL of lysis buffer). Cells containing the lysis buffer scraped into Eppendorf tubes and incubated on ice for 10 min to complete the lysis process. The samples were centrifuged for 15 min at 13,000 rpm, 4 °C. The supernatant was collected, and protein concentration of cell lysates were determined using Bradford assay (Bio-Rad). Cells treated with hypoxia were harvested in the hypoxia chamber to prevent reoxygenation. 20 μg of protein was prepared in $2 \times$ SDS loading buffer (100 mM Tris-HCl pH 6.8, 20% v/v glycerol, 4% w/v SDS, 200 mM DTT and Bromophenol Blue). SDS-PAGE and immunoblots were carried out using standard protocols. Images were obtained by chemiluminescence using ChemiDoc (Bio-Rad) or EcoMax X-Ray film processor (Protec). ImageJ was used for quantification of immunoblots with band intensities normalised to β-Actin.

# Data availability

The HeLa NF-κB depletion in response to hypoxia RNA-seq dataset is accessible at NCBI GEO: GSE260616. The SKNAS hypoxia stimulation RNA-seq dataset is accessible at NCBI GEO: GSE284326.

The source data of this paper are collected in the following database record: biostudies:S-SCDT-10_1038-S44319-025-00651-x.

# Peer review information

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

## Acknowledgements

We would like to acknowledge the members of Rocha and Kenneth labs for helpful discussions and the member of the hypoxia and NF-kappaB community for sharing their RNA-seq data. We thank Dr. Marco Sciacovelli (University of Liverpool) for providing the OxPhos antibody set and Dr. Dom Byrne for valuable reagents. We acknowledge the Liverpool Centre for Cell Imaging (CCI) for provision of imaging equipment, in particular the Epifluorescent microscope funded by MRC grant number MR/K015931/1, OMERO and technical assistance. This work was funded by a Wellcome trust collaborator grant to SR (206293/Z/17/Z) and the University of Liverpool, CSK is supported by a fellowship from the Royal College of Surgeons in England (RCSEng), a Tung Postgraduate fellowship and Alder Hey Children's Charity. NKS is supported by Worldwide Cancer Research [grant number 24-0353].

## Author contributions

**Dilem Shakir**: Conceptualization; Data curation; Formal analysis; Validation; Investigation; Visualization; Writing—original draft; Writing—review and editing. **Michael Batie**: Data curation; Formal analysis; Validation; Investigation; Visualization; Writing—original draft; Writing—review and editing. **Chun-Sui Kwok**: Formal analysis; Investigation. **Simon J Cook**: Resources; Writing—review and editing. **Niall S Kenneth**: Funding acquisition; Writing—original draft; Writing—review and editing. **Sonia Rocha**: Conceptualization; Formal analysis; Supervision; Funding acquisition; Investigation; Writing—original draft; Project administration; Writing—review and editing.

Source data underlying figure panels in this paper may have individual authorship assigned. Where available, figure panel/source data authorship is listed in the following database record: biostudies:S-SCDT-10_1038-S44319-025-00651-x.

## Disclosure and competing interests statement

The authors declare no competing interests.

