## [Peer Review File · EMBO Reports]

NF- κ B is a Central Regulator of Hypoxia-Induced Gene Expression

Dilem Shakir, Michael Batie, Chun-Sui Kwok, Simon J. Cook, Niall Kenneth, and Sonia Rocha

Corresponding author(s): Sonia Rocha (Sonia.Rocha@liverpool.ac.uk)

Review Timeline:

Transfer Date:	2nd Jun 25
Editorial Decision:	4th Jul 25
Revision Received:	20th Oct 25
Accepted:	13th Nov 25

Editor: Esther Schnapp

**Transaction Report: This manuscript was transferred to
EMBO reports following peer review at Review Commons.**

**Review
COMMONS**

Review #1

1. Evidence, reproducibility and clarity:

Evidence, reproducibility and clarity (Required)

The work from Shakir et al uses different cell line models to investigate the role of NF- κ B in the transcriptional adaptation of cells to hypoxia, which is relevant. In addition, the manuscript contains a large amount of data that could be of interest and even useful for researchers in the field of hypoxia and NF- κ B. However, in my opinion, there are several concerns that should be revised and additional experiments that could be included to strengthen the relevance of the work.

****Specific issues:****

In Figure 1A, the authors examine which of the genes induced by hypoxia require NF- κ B by RNA sequencing analysis of cells knocked down for specific NF- κ B subunits and exposed to hypoxia for 24 hours. The knockdown is about 40-60% at the RNA level, but it would be helpful to show the effect of knockdown at the protein level.

All the data regarding genes induced by hypoxia in control or NF- κ B siRNA-treated cells are somewhat confusing. If I understand correctly, when the data from the three different siRNAs are crossed, only 1070 genes are upregulated and 295 are downregulated in an NF- κ B-independent manner. If this is the case, I think it would be easier to use this information in Figure 2 to define the hypoxia-induced genes that are NF- κ B-dependent by simply considering those induced in the control that are not in the NF- κ B-independent subset (rather than repeating the integration of the data without additional explanation). If the authors do this simple analysis, are the resulting genes the same or similar? In any case, the way these numbers are obtained should be shown more clearly (i.e., a new Venn diagram showing genes up- or down-regulated in the siRNA control that are not up- or down-regulated in any of the siRNA-NF- κ B treatments).

Figure 2H shows that approximately 80% of the NF- κ B-dependent genes up- or down-regulated in hypoxia were identified as RelA targets, which is statistically significant compared to RelB or cRel targets. However, what is the proportion of genes identified as RelA targets in the subset of NF- κ B-independent hypoxia-induced genes? And in a randomly selected set of 500-600 genes? In my opinion, this statistical analysis should be included to demonstrate a relationship between NF- κ B recruitment and hypoxia-induced upregulation (expected) and downregulation (unexpected). In this context, it is surprising that HIF consensus sites are preferentially detected in the genes that are supposed to be NF- κ B dependent instead of RelA consensus.

Figure 3 is just a confirmation by qPCR of the data obtained in the RNA-seq analysis, which in my opinion should be included as supplementary information. Moreover, both the effects of hypoxia and reversion by RelB siRNA are modest in several of the genes tested. The same is true for Figures 4 and 5 with very modest and variable results across cell types and genes.

Figure 6 shows the effect of NF- κ B knockdown on the induction of ROS in response to hypoxia. In the images provided, the effect of hypoxia is minimal in control cells, with the only clear differences shown in RelA-depleted cells. In 6B it is not clear what the three asterisks in the normoxia control represent (compared to the hypoxia siRNA control?). This should be clarified in the figure legend or text. In the Western blot of 6C, there are no differences in the levels of SOD1 after RelA depletion. Again, there is no reason not to include the NF- κ B subunits in the Western blot analysis.

Finally, regarding Figure 7, the authors mention that "we confirmed that hypoxia led to a reduction in several proteins represented in this panel (of proteins involved in oxidative phosphorylation), such as UQCRC2 and IDH1 (Figure 7A-B)". The authors cannot say this because it is not seen in the Western blot in 7A or in the quantification shown in 7B. In my personal opinion, stating something that is not even suggested in the experiments is very negative for the credibility of the whole message.

In conclusion, this paper contains a large amount of relevant information, but i) non-essential data should be moved to Supplementary, ii) protein levels of relevant players need to be shown in addition to RNA, iii) minimal or undetectable differences need to be considered as no-differences, and iv) a model showing what is the interpretation of the data provided is needed to better understand the message of the paper. I mean, is it p65 or RelB binding to some of these genes leading to their activation or repression, or is it RelA or RelB inducing HIF1 β leading to NF- κ B-dependent gene activation by hypoxia? If this were the case, experimental evidence that NF- κ B regulates a subset of hypoxia genes through HIF1 β would make the story more understandable.

2. Significance:

Significance (Required)

The work presented here is interesting but does not provide a major advance over previous publications, the main message being that a subset of hypoxia-regulated genes are NF- κ B dependent. However, there is no mechanistic explanation of how this regulation is achieved and several data that are not clearly connected. A more comprehensive analysis of the data and additional experimental validation would greatly enhance the significance of the work.

3. How much time do you estimate the authors will need to complete the suggested revisions:

Estimated time to Complete Revisions (Required)

(Decision Recommendation)

Between 3 and 6 months

Yes

Review #2

1. Evidence, reproducibility and clarity:

Evidence, reproducibility and clarity (Required)

In this study, the authors have interrogated the role of NF-kappaB in the cellular transcriptional response to hypoxia. While HIF is considered the master regulator of the cellular response to hypoxia, it has long been known that multiple transcription factors also play a role both independently of HIF and through the regulation of HIF-1alpha levels. Chief amongst these is NF-kappaB, a regulator of cell death and inflammation amongst other things. While NF-kappaB has been known to be activated in hypoxia through altered PhD activity, the impact of this on global gene expression has remained unclear and this study addresses this important question. Of particular interest, genes downregulated in hypoxia appear to be repressed in a NF-kappaB-dependent manner. Overall, this nice study reveals an important role for NF-kappaB in the control of the global cellular transcriptional response to hypoxia.

2. Significance:

Significance (Required)

Some questions for the authors to consider with experiments or discussion:

- One caveat of the current study which should be discussed is that while interesting and extensive, the analysis is restricted to cancer cell lines which have dysfunctional gene expression systems which may differ from "normal" cells. This should be discussed.
- In the publicly available data sets analysed, were the same hypoxic conditions used as in this study. This information should be included.
- What is known about NF-kappaB as a transcriptional repressor in other systems such as the control of cytokine or infection driven inflammation? This is briefly discussed but should be expanded. This is important as a key question in the study of hypoxia is what regulates gene repression.
- NF-kappaB has previously been shown to regulate HIF-1alpha transcription. What are the effects of NF-kappaB subunit siRNAs on basal HIF-1alpha transcription? In figure 7, it appears that NF-kappaB subunit siRNA is without effect on hypoxia-induced HIF protein expression. Could this account for some of the effects of NF-kappaB depletion on the hypoxic gene signature? This point needs to be clarified in light of the data presented.
- NRF-2 is a key cellular sensor of oxidative stress in a similar way to HIF being a hypoxia sensor. The authors demonstrate using a dye that ROS are paradoxically increased in hypoxia (a more controversial finding than the authors present). It would be of interest to know if NRF-2 is induced in hypoxia as a marker of cellular oxidative stress. Similarly it would be interesting to determine by metabolic analysis whether oxidative phosphorylation (O₂ consumption) is decreased as the transcriptional signature would suggest (although the difficulty of performing metabolic analysis in hypoxia is acknowledged).

3. How much time do you estimate the authors will need to complete the suggested revisions:

Estimated time to Complete Revisions (Required)

(Decision Recommendation)

Between 1 and 3 months

4. Review Commons values the work of reviewers and encourages them to get credit for their work. Select 'Yes' below to register your reviewing activity at Web of Science Reviewer Recognition Service (formerly Publons); note that the content of your review will not be visible on Web of Science.

Yes

Review #3

1. Evidence, reproducibility and clarity:

Evidence, reproducibility and clarity (Required)

Strengths

This manuscript attempts to integrate multiple strands of data to determine the role of NFkB in hypoxia -induced gene expression. This analysis looks at multiple NFkB subunits in multiple cell lines to convincingly demonstrate that NFkB does indeed play a central role in the regulation of hypoxia-induced gene expression. This broad approach integrates new experimental data with findings from the published literature.

A significant amount of work has been performed both experimentally and bioinformatically to test experimental hypotheses.

Limitations

The main analysis in the paper involves comparing the impact of knocking down different NFkB family members in hypoxia and comparing transcriptional responses. I am surprised that the authors did not include the impact of knockdown of the NFkB family members in normoxia too. The absence of these control experiments allows us to understand the role of NFkB in hypoxia, but does not give us information as to how many of those impacts are specific/ induced in hypoxic conditions. i.e. many of the observed effects of NFkB knockdown could be due to basal suppression of NFkB target genes that happen to be hypoxia sensitive. This finding is obviously important, but it would be nice to know how many of those genes are only / preferentially regulated by NFkB in hypoxia. This would give a much deeper insight into the role of NFkB in hypoxia induced gene expression.

The broad experimental approach while a strength of the paper in many ways also has its limitations e.g. Motif analysis revealing e.g. HIF-1a binding site enrichment in RelA and RelB-dependent DEGs is correlative observation and does not prove HIF involvement in NFkB-dependent hypoxia induced gene activation. Comparing responses with responses seen in one cell type with responses that have been described in a database comprised of

many studies in a variety of different cells also has some limitations. These points can be described more fully in the discussion

For siRNA transfections, single oligonucleotide sequences were used for RelA, RelB and cRel. This increases the potential likelihood of 'off targets' compared to pooled oligos delivered at lower concentrations. This limitation should at least be mentioned.

RNA-seq experiments are performed on n=2 data which means relatively low statistical power. How has the statistical analysis been performed on normalised counts (corresponding to 2 n- numbers) to yield statistical significance? I am not familiar with hypergeometric tests - please justify their use here.

P14

RelB is described as having the most widespread impact of hypoxia dependent gene changes across all cell systems tested. Could this be due to a more potent silencing of RelB and / or due to particularly high/ low expression of RelB in these cells in general?

P18

For western blot analysis best practise is to have 2 MW markers per blot presented

For quantification, I suggest avoiding performing statistical analysis on semi-quantitative data unless a dynamic range of detection (with standards) has been fully established.

P19

There is clearly an effect of reciprocal silencing with the NFkB knockdown experiments ie. siRelA affects RelB levels in hypoxia and vice versa. The implications of this for data interpretation should be discussed.

P20

The literature can be better cited in relation RelB and hypoxia

A brief search reveals a few papers that should be mentioned/ discussed.

Oliver et al. 2009

Patel et al. 2017

Riedl et al. 2021

I suggest leaving out mention of I κ B α sumoylation and supplementary figure 10. I'm not sure the data in the paper as a whole merits focus on this very specific point.

There is a very strong reliance on mRNA and TPM data. Some additional protein data in support of key findings will enhance

A graphical abstract summarising key findings with exemplar genes highlighted will enhance.

Both HIF and NF κ B are ancient evolutionarily conserved pathways. Can lessons be learned from evolutionary biology as to how NF κ B regulation of hypoxia induced genes occurred. Does the HIF pathway pre-date the NF κ B pathway or vice versa. This approach could be valuable in supporting the findings from this study.

****Minor comments****

P2 please briefly explain how 5 genes give rise to 7 proteins

P2 there seems to be some recency bias in the studies cited as being associated with NF κ B activation in response to hypoxia. Mention of Koong et al (1994) and Taylor et al (1999) and other early papers in the field will enhance

P3

The role of PHD enzymes in the regulation of NF κ B in hypoxia can be introduced and / or discussed

P8

I suggest use of proportional Venn diagrams to demonstrate the patterns more clearly

P11 To what extent might NF κ B and Rest co-operate/ co-regulate gene repression in hypoxia?

2. Significance:

Significance (Required)

Shakir et al. present a manuscript titled 'NFkB is a central regulator of hypoxia-induced gene expression'.

The research group are experts in both NFkB and hypoxia signaling and are the ideal group to perform these studies.

Hypoxia and inflammation are co-incident in many physiological and pathophysiological conditions, where the microenvironment affects disease severity and patient outcome. The cross talk between inflammatory and hypoxia signaling pathways is not fully described. Thus, this manuscript takes a novel approach to an established question and concludes clearly that NFkB is a central regulator of hypoxia-induced gene expression.

3. How much time do you estimate the authors will need to complete the suggested revisions:

Estimated time to Complete Revisions (Required)

(Decision Recommendation)

Between 3 and 6 months

Yes

Full Revision

Manuscript number: RC-2025-02863

Corresponding author(s): Sonia, Rocha

[Please use this template only if the submitted manuscript should be considered by the affiliate journal as a full revision in response to the points raised by the reviewers.]

*If you wish to submit a preliminary revision with a revision plan, please use our "Revision Plan" template. **It is important to use the appropriate template to clearly inform the editors of your intentions.**]*

1. General Statements [optional]

We would like to thank all the reviewers for their positive comments and valuable feedback. In addition, we would like to address reviewer 1 query on novelty, which was not questioned by the other 2 reviewers. Our study uncovered two main aspects of hypoxia biology: first we addressed the role of NF-kappaB contribution towards the transcriptome changes in hypoxia, and second, this revealed a previously unknown aspect, that NF-kappaB is required for gene repression in hypoxia. While we know a lot about gene induction in hypoxia, much less is known about repression of genes. In times of energy preservation, gene repression is as important as gene induction.

.

This section is mandatory. Please insert a point-by-point reply describing the revisions that were already carried out and included in the transferred manuscript.

Reviewer #1 (Evidence, reproducibility and clarity (Required)):

The work from Shakir et al uses different cell line models to investigate the role of NF-kB in the transcriptional adaptation of cells to hypoxia, which is relevant. In addition, the manuscript contains a large amount of data that could be of interest and even useful for researchers in the field of hypoxia and NF-kB. However, in my opinion, there are several concerns that should be revised and additional experiments that could be included to strengthen the relevance of the work.

We thank this reviewer for their positive comments.

Specific issues:

In Figure 1A, the authors examine which of the genes induced by hypoxia require NF-kB by RNA sequencing analysis of cells knocked down for specific NF-kB subunits and exposed to hypoxia for

24 hours. The knockdown is about 40-60% at the RNA level, but it would be helpful to show the effect of knockdown at the protein level.

We agree with this and have added Western blot data (Sup. Figure S1F), which shows the effects of the siRNA are much more pronounced at the protein level.

All the data regarding genes induced by hypoxia in control or NF- κ B siRNA-treated cells are somewhat confusing. If I understand correctly, when the data from the three different siRNAs are crossed, only 1070 genes are upregulated and 295 are downregulated in an NF- κ B-independent manner. If this is the case, I think it would be easier to use this information in Figure 2 to define the hypoxia-induced genes that are NF- κ B-dependent by simply considering those induced in the control that are not in the NF- κ B-independent subset (rather than repeating the integration of the data without additional explanation). If the authors do this simple analysis, are the resulting genes the same or similar? In any case, the way these numbers are obtained should be shown more clearly (i.e., a new Venn diagram showing genes up- or down-regulated in the siRNA control that are not up- or down-regulated in any of the siRNA-NF- κ B treatments).

Figure 1 shows the effects on gene expression of hypoxia in control and NF- κ B subunit-depleted cells compared to normoxia control cells. Figures 1F/1G compares genes up/downregulated in hypoxia when RelA, RelB, and cRel are depleted, compared to normoxia control. Figure 1 does not display NF- κ B-dependent/independent hypoxia-responsive genes, but rather the overall effect of siRNA control and siNF- κ B treatments in hypoxia, compared to siRNA control in normoxia. Figure 2 then defines NF- κ B-dependent and independent hypoxia-responsive genes. We actually define these exactly as the reviewer suggested and agree that we should show the way these numbers are obtained more clearly. We have added the suggested Venn diagrams (Sup. Figure S2) and added extra information to the methods section (page 5 of revised manuscript). We felt it was important to show all the data upfront in Figure 1 and then integrate and focus on NF- κ B-dependent hypoxia-induced genes in Figure 2.

Figure 2H shows that approximately 80% of the NF- κ B-dependent genes up- or down-regulated in hypoxia were identified as RelA targets, which is statistically significant compared to RelB or cRel targets. However, what is the proportion of genes identified as RelA targets in the subset of NF- κ B-independent hypoxia-induced genes? And in a randomly selected set of 500-600 genes? In my opinion, this statistical analysis should be included to demonstrate a relationship between NF- κ B recruitment and hypoxia-induced upregulation (expected) and downregulation (unexpected). In this context, it is surprising that HIF consensus sites are preferentially detected in the genes that are supposed to be NF- κ B dependent instead of RelA consensus.

We thank the reviewer for this question, which is really helpful. The way we have displayed the stars on the graph for Figure 2H was slightly misleading we realize now. As such, we have

amended the graph. RelA, RelB, and cRel bound genes (from the ChIP atlas) are all significantly enriched within our NF- κ B-dependent hypoxia-responsive genes, there is no statistical testing between RelA bound vs RelB bound or cRel bound. We have also performed this analysis on the NF- κ B-independent hypoxia-responsive genes and see the same trend (Sup. Figure S5B). This indicates that the enrichment of Rel binding sites from the ChIP atlas is not specific to NF- κ B-dependent hypoxia-responsive genes. We have moved Figure 2H to (Sup. Figure S5A) and amended our description of the finding. This showcases how DNA binding does not necessarily mean functionality. We have amended our description of this result and limitation of the study.

Figure 3 is just a confirmation by qPCR of the data obtained in the RNA-seq analysis, which in my opinion should be included as supplementary information. Moreover, both the effects of hypoxia and reversion by RelB siRNA are modest in several of the genes tested. The same is true for Figures 4 and 5 with very modest and variable results across cell types and genes.

We appreciate this comment; we would like to keep this as a main figure for full transparency and show validation of our RNA-sequencing results.

Figure 6 shows the effect of NF- κ B knockdown on the induction of ROS in response to hypoxia. In the images provided, the effect of hypoxia is minimal in control cells, with the only clear differences shown in RelA-depleted cells.

The quantification of the IF data (Figure 6B) shows ROS induction in hypoxia which is reduced in Rel-depleted cells, with RelA depletion having the strongest effect. ROS generation in hypoxia, although counterintuitive, is well documented and used for important signalling events. We believe our data supports the previously reported levels of ROS induction (reviewed in {Alva, 2024}) in hypoxia and importantly, that NF- κ B depletion can at least partially reverse this.

In 6B it is not clear what the three asterisks in the normoxia control represent (compared to the hypoxia siRNA control?). This should be clarified in the figure legend or text.

We apologize for the lack of clarity we have now added this information to the figure legend.

In the Western blot of 6C, there are no differences in the levels of SOD1 after RelA depletion. Again, there is no reason not to include the NF- κ B subunits in the Western blot analysis.

We have added the Western blot analysis to this figure. We were trying to simplify it. Although depletion of RelA does not rescue the hypoxia-induced repression of SOD1, depletion of RelB

does. Furthermore, cRel although not statistically significant, has a trend for the rescue of this effect, see Figure 6C-D.

Finally, regarding Figure 7, the authors mention that "we confirmed that hypoxia led to a reduction in several proteins represented in this panel (of proteins involved in oxidative phosphorylation), such as UQCRC2 and IDH1 (Figure 7A-B)". The authors cannot say this because it is not seen in the Western blot in 7A or in the quantification shown in 7B. In my personal opinion, stating something that is not even suggested in the experiments is very negative for the credibility of the whole message.

We really do not agree with this comment. We do see reductions in the levels of the proteins we mentioned. We have made the figure less complex given that some proteins are very abundant while others are not. We hope the changes are now clear and apparent. We have changed the quantification normalisation to reflect this as well and modified our description of the results, see Figure 7 and Sup. Figure S18.

In conclusion, this paper contains a large amount of relevant information, but i) non-essential data should be moved to Supplementary, ii) protein levels of relevant players need to be shown in addition to RNA, iii) minimal or undetectable differences need to be considered as no-differences, and iv) a model showing what is the interpretation of the data provided is needed to better understand the message of the paper. I mean, is it p65 or RelB binding to some of these genes leading to their activation or repression, or is it RelA or RelB inducing HIF1beta leading to NF-kB-dependent gene activation by hypoxia? If this were the case, experimental evidence that NF-kB regulates a subset of hypoxia genes through HIF1beta would make the story more understandable.

We apologise but we do not know why the reviewer mentions HIF1beta. For gene induction, there is cooperation with the HIF system in some genes but not all. The most interesting and unexpected finding is that NF-kappaB is required for gene repression in hypoxia. We have added a new figure, investigating how HDAC inhibition could reverse the repression. A mechanism known to be employed by NF-kappaB when repressing genes. We have added all the blots for NF-kB, clarified the quantification and included other approaches including a CRISPR KO cell lines for both IKKs. We hope this is now clear.

Reviewer #1 (Significance (Required)):

The work presented here is interesting but does not provide a major advance over previous publications, the main message being that a subset of hypoxia-regulated genes are NF-kB dependent. However, there is no mechanistic explanation of how this regulation is achieved and several data that are not clearly connected. A more comprehensive analysis of the data and

additional experimental validation would greatly enhance the significance of the work.

We politely disagree with the reviewer. Our main finding is that NF- κ B does play an important role in gene regulation in hypoxia but unexpectedly, this occurs mostly via gene repression. While there is vast knowledge on gene induction in hypoxia, gene repression, which typically does not occur directly via HIF, is virtually unknown. A previous study had identified Rest as a transcriptional repressor {PMID: 27531581} but this could only account for 20% of gene repression. Our findings reveal up to 60% of genes repressed in hypoxia require NF- κ B, hence this is a significant finding and a major advance over previous knowledge. Furthermore, we feel this paper is an excellent data resource for the field, as it is, to our knowledge, the first study characterising the extent to which NF- κ B is required for hypoxia-induced gene changes, on a transcriptome-wide scale. Furthermore, we have validated this across multiple cell types and also used different approaches to investigate the role of NF- κ B in the hypoxia transcriptional response. We are happy that the other reviewers agree with our novel findings.

Reviewer #2 (Evidence, reproducibility and clarity (Required)):

In this study, the authors have interrogated the role of NF-kappaB in the cellular transcriptional response to hypoxia. While HIF is considered the master regulator of the cellular response to hypoxia, it has long been known that multiple transcription factors also play a role both independently of HIF and through the regulation of HIF-1alpha levels. Chief amongst these is NF-kappaB, a regulator of cell death and inflammation amongst other things. While NF-kappaB has been known to be activated in hypoxia through altered Phd activity, the impact of this on global gene expression has remained unclear and this study addresses this important question. Of particular interest, genes downregulated in hypoxia appear to be repressed in a NF-kappaB-dependent manner. Overall, this nice study reveals an important role for NF-kappaB in the control of the global cellular transcriptional response to hypoxia.

We thank this reviewer for their positive comments.

Reviewer #2 (Significance (Required)):

Some questions for the authors to consider with experiments or discussion:

-One caveat of the current study which should be discussed is that while interesting and extensive, the analysis is restricted to cancer cell lines which have dysfunctional gene expression systems which may differ from "normal" cells. This should be discussed.

We thank the reviewer for these comments. This is indeed an important aspect, which we now expand on in the discussion section. We also took advantage of RNA-seq datasets for HUVECs (a non-transformed cell lines) in response to hypoxia (Sup. Figure S15), TNF-alpha with and

without RelA depletion (Sup. Figure S16). These data support our findings that in hypoxia NF- κ B is important for transcriptional repression, with some contributions to gene induction, even in a non-transformed cell system.

In the publicly available data sets analyzed, were the same hypoxic conditions used as in this study. This information should be included.

We apologize if this was not clear, the hypoxia RNA-seq studies are the same oxygen level and time (1%, 24 hours), this is in the legend of Figure 4A and Sup. Figure S9 and in Sup. Table S2. We have added this information to the main text also.

- What is known about NF-kappaB as a transcriptional repressor in other systems such as the control of cytokine or infection driven inflammation? This is briefly discussed but should be expanded. This is important as a key question in the study of hypoxia is what regulates gene repression.

We have included this in the discussion and also analysed available data in HUVECs in response to cytokine stimulation with and without RelA depletion (Sup. Figure S16). This analysis revealed equal importance of RelA for activation and repression of genes upon TNF-alpha stimulation. Around 40% of genes require RelA for their induction or repression in response to TNF-a. In the discussion we have also included other references where NF-kappaB has been found to repress genes.

NF-kappaB has previously been shown to regulate HIF-1alpha transcription. What are the effects of NF-kappaB subunit siRNAs on basal HIF-1alpha transcription? In figure 7, it appears that NF-kappaB subunit siRNA is without effect on hypoxia-induced HIF protein expression. Could this account for some of the effects of NF-kappaB depletion on the hypoxic gene signature? This point needs to be clarified in light of the data presented.

We have included data for HIF-1 α RNA levels in HeLa cells with/without NF- κ B depletion followed by 24 hours of hypoxia (Sup. Figure S20) and we see a small reduction (~10-20%). The reviewer is correct, there was not much effect of NF- κ B depletion on HIF-1 α protein levels following 24 hours hypoxia in HeLa cells. Effects of NF-kappaB depletion can be found usually with lower times of hypoxia exposure or when more than one subunit is depleted at the same time. We have added this as a discussion point in the revised manuscript.

NRF-2 is a key cellular sensor of oxidative stress in a similar way to HIF being a hypoxia sensor. The authors demonstrate using a dye that ROS are paradoxically increased in hypoxia (a more

controversial finding than the authors present). It would be of interest to know if NRF-2 is induced in hypoxia as a marker of cellular oxidative stress. Similarly, it would be interesting to determine by metabolic analysis whether oxidative phosphorylation (O₂ consumption) is decreased as the transcriptional signature would suggest (although the difficulty of performing metabolic analysis in hypoxia is acknowledged).

To investigate if NRF2 is induced, we performed a western blot at 0, 1, and 24 hours 1% oxygen, but didn't see any induction of NRF2 protein levels (Sup. Figure S17A). We also overlapped our hypoxia upregulated genes with NRF2 target genes from {PMID:24647116 and PMID: 38643749} (Sup. Figure S17B) and found limited evidence of NRF2 target genes being induced. Based on these findings, it seems that NRF2 is not being induced in hypoxia, at least not at the hypoxia level/time point we have analysed. We also agree it would be ideal to measure oxygen consumption in hypoxia, but unfortunately, we do not have the technical ability to do this at present.

Reviewer #3 (Evidence, reproducibility and clarity (Required)):

Strengths

This manuscript attempts to integrate multiple strands of data to determine the role of NFκB in hypoxia-induced gene expression. This analysis looks at multiple NFκB subunits in multiple cell lines to convincingly demonstrate that NFκB does indeed play a central role in the regulation of hypoxia-induced gene expression. This broad approach integrates new experimental data with findings from the published literature.

A significant amount of work has been performed both experimentally and bioinformatically to test experimental hypotheses.

We thank this reviewer for their positive comments.

Limitations

The main analysis in the paper involves comparing the impact of knocking down different NFκB family members in hypoxia and comparing transcriptional responses. I am surprised that the authors did not include the impact of knockdown of the NFκB family members in normoxia too. The absence of these control experiments allows us to understand the role of NFκB in hypoxia, but does not give us information as to how many of those impacts are specific/ induced in hypoxic conditions. i.e. many of the observed effects of NFκB knockdown could be due to basal suppression of NFκB target genes that happen to be hypoxia sensitive. This finding is obviously important, but it would be nice to know how many of those genes are only / preferentially regulated

by NFkB in hypoxia. This would give a much deeper insight into the role of NFkB in hypoxia induced gene expression.

We agree this would have been ideal. For financial reasons we limited our analysis to hypoxia samples. We have performed qPCR analysis depleting RelA, RelB and cRel under normal oxygen conditions in HeLa (Sup. Figure S8). We find that the majority of the validated genes in HeLa cells which require NF- κ B for gene changes in hypoxia, are not regulated by NF- κ B under normal oxygen conditions. We have also added this limitation into our discussion section.

The broad experimental approach while a strength of the paper in many ways also has its limitations e.g. Motif analysis revealing e.g. HIF-1a binding site enrichment in RelA and RelB-dependent DEGs is correlative observation and does not prove HIF involvement in NFkB-dependent hypoxia induced gene activation. Comparing responses with responses seen in one cell type with responses that have been described in a database comprised of many studies in a variety of different cells also has some limitations. These points can be described more fully in the discussion

We agree these are mere correlations and hence a limitation and we have not formerly tested the involvement of HIF. We have included this in the discussion as suggested. For HIF binding site correlation, we do also compare to HIF ChIP-seq in HeLa cells exposed to 1% oxygen, albeit at 8 hours and not 24 hours (Sup. Figure S4).

For siRNA transfections, single oligonucleotide sequences were used for RelA, RelB and cRel. This increases the potential likelihood of 'off targets' compared to pooled oligos delivered at lower concentrations. This limitation should at least be mentioned.

We agree and have now included this as a limitation in the discussion section. We have now also included analysis using wild type and 2 different IKK α / β double KO CRISPR cell lines generated in the following publication {PMID: 35029639}. Out of the 9 genes we identified as NF- κ B-dependent hypoxia upregulated genes from HeLa cell RNA-seq and validated by qPCR, which are also hypoxia-responsive in HCT116 cells (Sup. Figure S11D), 6 displayed NF- κ B dependence in HCT116 cells (Sup. Figure S14). We also provide new protein data in this cell system for oxidative phosphorylation markers, which show as with the siRNA depletion, rescue of repression of these proteins when NF- κ B is inactivated.

RNA-seq experiments are performed on n=2 data which means relatively low statistical power. How has the statistical analysis been performed on normalised counts (corresponding to 2 n-numbers) to yield statistical significance? I am not familiar with hypergeometric tests - please justify their use here.

We use DESeq2 for differential expression analysis and filter for effect size ($> -/+ 0.58 \log_2$ fold change) and statistical significance ($FDR < 0.05$) to determine differentially expressed genes (DEGs). DESeq2 can use a minimum of 2 biological replicates. In the DESeq2 analysis, raw gene counts are converted to DESeq2 normalised counts which are then used for differential expression analysis.

I am not familiar with hypergeometric tests - please justify their use here.

The hypergeometric test (equivalent to a one-sided Fisher's exact test) is routinely used to determine whether the observed overlap between two gene lists is statistically significant compared to what would be expected by chance. It is also the statistical test of choice for popular bioinformatics tools which perform over representation analysis (ORA) to see which gene sets/groups/pathways/ontologies are over-represented in a gene list, examples include Metascape, clusterProfiler, WebGestalt (used in this study), and gProfiler.

P14

RelB is described as having the most widespread impact of hypoxia dependent gene changes across all cell systems tested. Could this be due to a more potent silencing of RelB and / or due to particularly high/ low expression of RelB in these cells in general?

This is an excellent point, at the RNA level the RelB depletion is slightly more efficient (Sup. Figure S1), at the protein level, silencing is highly potent with all 3 siRNAs (Sup. Figure S1). We looked at the RNA levels of RelA, RelB and cRel in HeLa cells at basal conditions, and RelA shows the highest abundance compared to RelB and cRel, while RelB and cRel have similar expression levels (see below). However, RelB is very dynamic in response to hypoxia, something we have observed but have not published yet.

Full Revision

P18

For western blot analysis best practise is to have 2 MW markers per blot presented

We have and have added the second MW markers suggested.

For quantification, I suggest avoiding performing statistical analysis on semi-quantitative data unless a dynamic range of detection (with standards) has been fully established.

We agree this has many limitations, we will keep the quantification but moved into supplementary information.

P19

There is clearly an effect of reciprocal silencing with the NFkB knockdown experiments ie. siRelA affects RelB levels in hypoxia and vice versa. The implications of this for data interpretation should be discussed.

Indeed, it is well known that RelB and cRel are RelA targets. Less is known about RelA as it is not a known NF- κ B target. We have added a discussion in the revised manuscript.

P20

The literature can be better cited in relation RelB and hypoxia

A brief search reveals a few papers that should be mentioned/ discussed.

Oliver et al. 2009

Patel et al. 2017

Riedl et al. 2021

We have looked into these suggestions. Oliver et al, refer to hypercapnia, not hypoxia and the other two only briefly mentioned RelB with no effects toward the goals of their studies. We have tried to incorporate what is currently known as much as possible.

I suggest leaving out mention of I κ B α sumoylation and supplementary figure 10. I'm not sure the data in the paper as a whole merits focus on this very specific point.

We thank the reviewer for this suggestion and we have removed this aspect from the manuscript.

There is a very strong reliance on mRNA and TPM data. Some additional protein data in support of key findings will enhance

We have added additional protein level analysis where we could obtain antibodies, see Figures 6, 7 and Sup. Figures S17, S18, and S19 for our protein level analysis.

A graphical abstract summarising key findings with exemplar genes highlighted will enhance.

We have added a model to summarise our findings as suggested.

Both HIF and NFkB are ancient evolutionarily conserved pathways. Can lessons be learned from evolutionary biology as to how NFkB regulation of hypoxia induced genes occurred. Does the HIF pathway pre-date the NFkB pathway or vice versa. This approach could be valuable in supporting the findings from this study.

We have investigated this. Unfortunately, there are very little available data on hypoxia gene expression in lower organisms. However, we have added a few sentences on the evolution of NF- κ B and HIF.

Minor comments

P2 please briefly explain how 5 genes give rise to 7 proteins

We have added this to the introduction as requested.

P2 there seems to be some recency bias in the studies cited as being associated with NFkB activation in response to hypoxia. Mention of Koong et al (1994) and Taylor et al (1999) and other early papers in the field will enhance

We have added these as suggested.

P3

The role of PHD enzymes in the regulation of NFkB in hypoxia can be introduced and / or discussed

We have added a reference to this aspect as suggested.

P8

I suggest use of proportional Venn diagrams to demonstrate the patterns more clearly

We have added these as suggested.

P11 To what extent might NFkB and Rest co-operate/ co-regulate gene repression in hypoxia?

This is a good question. We have overlapped our datasets with Rest-dependent hypoxia-regulated genes identified by Cavadas et al., (Figure below), and find that these appear to act independently of each other for the most part, with very few genes co-regulated by both.

Reviewer #3 (Significance (Required)):

Shakir et al. present a manuscript titled 'NFκB is a central regulator of hypoxia-induced gene expression'.

The research group are experts in both NFκB and hypoxia signaling and are the ideal group to perform these studies.

Hypoxia and inflammation are co-incident in many physiological and pathophysiological conditions, where the microenvironment affects disease severity and patient outcome. The cross talk between inflammatory and hypoxia signaling pathways is not fully described. Thus, this manuscript takes a novel approach to an established question and concludes clearly that NFκB is a central regulator of hypoxia-induced gene expression.

We thank the reviewer for these positive comments.

Dear Prof. Rocha,

Thank you for the submission of your revised manuscript to EMBO reports. We have now received all referee reports on it, as well as referee cross-comments that are pasted below.

As you will see, all referees acknowledge that the ms has been improved during revision. Referees 1 and 2 (former referees 2 and 3) are overall satisfied with the revised ms, however, referee 3 (former referee 1) still has remaining concerns and the main concern is that gene expression changes after NF- κ B knockdown should be compared for each cellular model under normoxic and hypoxic conditions side-by-side. I included your reply to this point when I asked the referees for cross-comments. Nevertheless, referee 1 agrees with referee 3 that this is an important point.

However, we are aware that not all experiments can be repeated now, and both referees 3 and 1 suggest that you could validate a panel of candidate NF- κ B-dependent hypoxia-responsive genes by qPCR, including normoxia-hypoxia conditions in the different cellular models. I think this is a good compromise, but please let me know if you disagree and we can discuss this further. All other remaining concerns should also be addressed.

In addition to the referee comments, some editorial requests will need to be addressed also. Please provide a point-by-point response to all comments with your final ms.

- Please submit the final ms as a Word file without figures. All main figures need to be uploaded as individual production quality Figure files.
- Please correct the conflict of interest subheading to "Disclosure and Competing Interests Statement"
- Please remove the author credits from the ms file. All credits need to be entered during online ms submission.
- We need a completed author checklist from you, which you can download from our author guidelines <<https://www.embopress.org/page/journal/14693178/authorguide>>. The completed author checklist will also be part of the transparent peer-review process file.
- The following funders are acknowledged in the ms file but not entered in the online submission system: MRC grant number MR/K015931/1, OMERO, the University of Liverpool. All funding information needs to be entered in both places.
- Figure panels 7A-C are called out before Figure 4; callouts for Figure 5A and Figure S7 are missing, please correct and add the callouts.
- 4 DATASETs are uploaded but the nomenclature needs to be corrected in all places (dataset files and ms file) to Dataset EV1-EV4.
- The supplemental material needs to be renamed to APPENDIX and the nomenclature in all places needs to be Appendix Figure S1-S20, Appendix Table S1-S3. We also need here a title page in the Appendix file with a table of content that has page numbers.
- The Methods section should include a Reagents and Tools Table (listing key reagents, experimental models, software and relevant equipment and including their sources and relevant identifiers) and a Methods and Protocols section in which we encourage authors to describe their methods using a step-by-step protocol format with bullet points, to facilitate the adoption of the methodologies across labs. More information on how to adhere to this format as well as downloadable templates (.docx) for the Reagents and Tools Table can be found in our author guidelines: <<https://www.embopress.org/page/journal/14693178/authorguide#manuscriptpreparation>>.
- The manuscript sections should be in the following order: Title page - Abstract & Keywords - Introduction - Results - Discussion - Methods - Data Availability - Acknowledgments - Disclosure Statement & Competing Interests - References - Figure Legends - (Main Tables with legends if applicable)

* Figure Legends - Comments *

- Please note that the exact p values are not provided in the legends of figures 3B-F, 5B-L; 6B, D. Please provide exact p-values as reasonable.
- Please note that information related to n is missing in the legends of figures 1B, C, D, E; 4B, this needs to be added.

EMBO press papers are accompanied online by A) a short (1-2 sentences) summary of the findings and their significance, B) 2-3 bullet points highlighting key results and C) a synopsis image that is exactly 550 pixels wide and 200-600 pixels high (the height is variable). The synopsis image should provide a sketch of the major findings, like a graphical abstract. Please note that text needs to be readable at the final size. Please send us this information along with the final manuscript.

I look forward to seeing a final version of your manuscript when it is ready.

Referee #1:

The authors have satisfactorily addressed my comments

Referee #2:

Shakir et al. present a manuscript titled 'NFkB is a central regulator of hypoxia-induced gene expression'.

The research group are experts in both NFkB and hypoxia signaling and are the ideal group to perform these studies. Hypoxia and inflammation are co-incident in many physiological and pathophysiological conditions, where the microenvironment affects disease severity and patient outcome. The crosstalk between inflammatory and hypoxia signaling pathways is not fully described. Thus, this manuscript takes a novel approach to an established question and concludes clearly that NFkB is

a central regulator of hypoxia-induced gene expression. This manuscript can be a valuable resource to the hypoxia and NFkB communities.

The manuscript submitted to EMBO reports has improved significantly since the pre-print stage and the authors have written a comprehensive and detailed rebuttal to my comments.

I only have minor outstanding comments.

1. In relation to the comment on limiting RNA-seq analysis to hypoxia samples

These experiments are welcome and support the idea that the majority of the validated genes are not regulated by NFkB in normoxia (however, it would have been nice if the qPCR experiments were performed under both normoxia and hypoxia (in the same experiment)).

2. In relation to the comment about statistical analysis of semi-quantitative western blot data.

I agree with putting the quantification in the supplemental data but the comment on statistical analysis stands.

I suggest keeping the quantification (in the supplemental data) and removing all of the statistical analysis of western blots.

3. In relation to the inclusion of more literature relating to RelB and hypoxia

Oliver et al. 2009 pmid: 19422287 is titled: Hypoxia Activates NF- κ B-Dependent Gene Expression Through the Canonical Signaling Pathway. There are several experiments directed to hypoxia and NFkB family members. This paper should likely also be cited.

The inclusion of the other two papers is welcome.

4.

In relation to proportional venn diagrams.

There are other venn diagrams that might also benefit from being presented in a proportional way (for consistency and

readability)

Referee #3:

This manuscript investigates the functional contribution of NF- κ B to the transcriptional regulation imposed by hypoxia. The authors demonstrate the existence of a core signature that is hypoxia-induced and NF- κ B-dependent across multiple cell types. Moreover, they propose that ROS generation in response to hypoxia depends on NF- κ B. These findings could be relevant for researchers in the NF- κ B field and beyond.

However, I have several concerns regarding the clarity of the presentation, the rigor of the experimental controls, and the strength of the conclusions.

Major concerns:

-Normalization and control comparisons:

The most relevant concern is about the definition of hypoxia-induced NF- κ B-dependent genes that are defined in figures 1 and 2. Specifically, Figure 1 shows that all comparisons are made relative to the untreated control siRNA. However, the initial differences between si-NF- κ B cells and control siRNA are not considered. Consequently, it is possible that genes classified as downregulated in KD NF- κ B cells are in fact comparably induced by hypoxia (not hypoxia-induced in an NF- κ B-dependent manner), and the conclusions taken are primarily due to the analysis approach. To avoid this confounding effect, comparing each cellular model under normoxic and hypoxic conditions is crucial.

-Clarity of figures and rationale:

The presentation of results and the accompanying text is often difficult to follow. For example, the Venn diagrams in Figure 1F are very confusing, and the sentence "When we compared the effect of individual NF- κ B subunits in hypoxia inducible gene signatures, all three subunits showed a shared proportion of genes between each other" is not clear.

Furthermore, in Figure 2D and 2E, it is difficult to reconcile the reported numbers with those in Figures 1F and 1G. The relationship between these analyses needs to be explicitly described, ideally with a clear schematic or explanatory text. Maybe because of my lack of background in the field, but I spent considerable time trying to understand Figures 1 and 2 without success, which underscores the need to improve clarity for readers who are not NF- κ B specialists.

-Inclusion of appropriate controls:

In Figure 3, normoxic controls should be included for all models to ensure that reduced induction upon hypoxia is not simply a reflection of lower basal expression levels (similar to that mentioned for Figure 1). In other words, the normoxia/hypoxia comparison for each condition should be shown and supported by appropriate statistical analysis.

-Quantification and statistical rigor:

The manuscript generally lacks quantitative data presentation. For example, in Figure S3, the identification of HIF1 binding motifs in RELA, RELB, or c-REL-dependent genes should be compared to randomly selected hypoxia-induced genes to assess enrichment significance.

In Figures 7A-C, the reductions in protein levels and their reversal upon NF- κ B depletion are, at best, marginal. The sentence stating that "Hypoxia led to reductions in several proteins represented in this panel..." is only partially supported by the data (perhaps only for IDH1). Similarly, differences in Vimentin and DYRK levels are negligible. Quantification with individual replicate data points should be provided for all experiments.

General comment:

Overall, the manuscript requires substantial revision of both text and figures to improve clarity and to present conclusions that are rigorously supported by the data. Including appropriate controls and clear normalization strategies is essential. At this stage, in my view, the manuscript does not provide sufficient experimental evidence to support several of the main claims. Additionally, the presentation makes it unnecessarily difficult for the general scientific community to distinguish between robust findings and observations that could be due to experimental variability.

I recommend that the authors carefully revise the whole manuscript to include all necessary controls, improve figure clarity, provide detailed quantifications, and rephrase conclusions where needed to reflect the actual strength of the data presented.

Cross-comments from referee 1:

In principle, I agree with the reviewer on the importance of normoxic controls. However, the authors may not be in a position to repeat all of the high throughput experiments with normoxic stimuli as this could be inordinately expensive. In this case, the authors could confirm some of their key genes in isolation and explain clearly in the text the limitations of not including normoxic samples. I still believe the paper is worthy of publication.

Cross-comments from referee 2:

In my opinion this experiment is not essential but is the optimal way to address the question. The authors have already provided some evidence that not all genes affected by NFKB in hypoxia are affected by silencing of NFKB family members in normoxia which in my view is the main point. Doing further experiments is unlikely to change this main finding.

Cross-comments from referee 3:

I think it is reasonable to request validation of a panel of candidate NF- κ B-dependent hypoxia-responsive genes by qPCR, including a normoxia-hypoxia conditions in the different cellular models, as suggested by reviewer 2. I still stand by my other comments about over-interpretation and a lack of clarity in several of the experiments. Could you please ask the authors to revise the text in general to improve the connection between what is observed and what is expressed?

Shakir et al_ NFkappaB is a central regulator of hypoxia-induced gene expression

We thank the reviewers for their detailed review and constructive feedback on our manuscript. They have highlighted some important points to improve the manuscript and some limitations, which we now discuss. They have also helped us better describe our work in the context of previous work from the field. We have addressed the comments below.

Referee #1:

The authors have satisfactorily addressed my comments

Referee #2:

Shakir et al. present a manuscript titled 'NFkB is a central regulator of hypoxia-induced gene expression'. The research group are experts in both NFkB and hypoxia signaling and are the ideal group to perform these studies. Hypoxia and inflammation are co-incident in many physiological and pathophysiological conditions, where the microenvironment affects disease severity and patient outcome. The crosstalk between inflammatory and hypoxia signaling pathways is not fully described. Thus, this manuscript takes a novel approach to an established question and concludes clearly that NFkB is a central regulator of hypoxia-induced gene expression. This manuscript can be a valuable resource to the hypoxia and NFkB communities.

The manuscript submitted to EMBO reports has improved significantly since the pre-print stage and the authors have written a comprehensive and detailed rebuttal to my comments.

I only have minor outstanding comments.

1. In relation to the comment on limiting RNA-seq analysis to hypoxia samples

These experiments are welcome and support the idea that the majority of the validated genes are not regulated by NFkB in normoxia (however, it would have been nice if the qPCR experiments were performed under both normoxia and hypoxia (in the same experiment)).

We agree and have performed qPCR experiments in both normoxia and hypoxia at the same time in HeLa cells and A549 cells with/without NF- κ B depletion, and HCT116 cells with/without IKK α/β double knockout (Appendix Figure S13-S17). In HeLa cells, we analysed 6 genes, each of these genes displayed NF- κ B-dependent control of RNA levels in hypoxia, but not in normoxia (Appendix Figure S14A-C). In A549 cells, we analysed 5 genes, each of these genes displayed NF- κ B dependent control of RNA levels in hypoxia, but not in normoxia. In HCT116 cells, 4 out of 6 genes tested, displayed IKK α/β dependence for hypoxia induction, and IKK α/β -dependent control of gene expression was specific to hypoxia. These data indicate, at least in the specific genes tested, that NF- κ B-dependent regulation of hypoxia responsive genes, occurs under hypoxia but not basal oxygen levels. However, further RNA-sequencing experiments with depletion or inhibition of NF- κ B subunits in

normoxia and hypoxia is required to properly delineate hypoxia specific effects. This limitation is mentioned in the discussion section.

2. In relation to the comment about statistical analysis of semi-quantitative western blot data.

I agree with putting the quantification in the supplemental data but the comment on statistical analysis stands. I suggest keeping the quantification (in the supplemental data) and removing all of the statistical analysis of western blots.

As suggested, we have moved the Western blot quantification to supplementary and removed the statistical analysis for Western blot quantification.

3. In relation to the inclusion of more literature relating to RelB and hypoxia Oliver et al. 2009 pmid: 19422287 is titled: Hypoxia Activates NF- κ B-Dependent Gene Expression Through the Canonical Signaling Pathway. There are several experiments directed to hypoxia and NF κ B family members. This paper should likely also be cited.

The inclusion of the other two papers is welcome.

We have now added more about the specific manuscript in the discussion.

4.
In relation to proportional venn diagrams.

There are other venn diagrams that might also benefit from being presented in a proportional way (for consistency and readability)

All Venn diagrams are now proportional.

Referee #3:

This manuscript investigates the functional contribution of NF- κ B to the transcriptional regulation imposed by hypoxia. The authors demonstrate the existence of a core signature that is hypoxia-induced and NF- κ B-dependent across multiple cell types. Moreover, they propose that ROS generation in response to hypoxia depends on NF- κ B. These findings could be relevant for researchers in the NF- κ B field and beyond.

However, I have several concerns regarding the clarity of the presentation, the rigor of the experimental controls, and the strength of the conclusions.

Major concerns:

-Normalization and control comparisons:

The most relevant concern is about the definition of hypoxia-induced NF- κ B-dependent genes that are defined in figures 1 and 2. Specifically, Figure 1 shows that all comparisons are made relative to the untreated control siRNA. However, the initial differences between si-NF- κ B cells and control siRNA are not considered. Consequently, it is possible that genes classified as downregulated in KD NF- κ B cells are in fact comparably induced by hypoxia (not hypoxia-induced in an NF- κ B-

dependent manner), and the conclusions taken are primarily due to the analysis approach. To avoid this confounding effect, comparing each cellular model under normoxic and hypoxic conditions is crucial.

Regarding the definition of hypoxia induced NF- κ B-dependent genes, please see below comments. We have performed qPCR analysis in normoxia and hypoxia with NF- κ B subunits depleted, please see referee #2 comment 1 response.

-Clarity of figures and rationale:

The presentation of results and the accompanying text is often difficult to follow. For example, the Venn diagrams in Figure 1F are very confusing, and the sentence "When we compared the effect of individual NF- κ B subunits in hypoxia inducible gene signatures, all three subunits showed a shared proportion of genes between each other" is not clear.

We have amended the text regarding Figures 1F and 1G. Also, we changed the labelling of the figure for more clear illustration.

Furthermore, in Figure 2D and 2E, it is difficult to reconcile the reported numbers with those in Figures 1F and 1G.

We identify NF- κ B-dependent hypoxia responsive genes in Figure 2 and Appendix Figure S2. The numbers in Figure 2D and 2E are reconciled by comparing to Appendix Figure S2. Also see the below comment response.

The relationship between these analyses needs to be explicitly described, ideally with a clear schematic or explanatory text. Maybe because of my lack of background in the field, but I spent considerable time trying to understand Figures 1 and 2 without success, which underscores the need to improve clarity for readers who are not NF- κ B specialists.

Figure 1 is comparing siControl, siRelA, siRelB and sicRel cells in hypoxia, with siControl cells in normoxia. The purpose of this figure is to show the overall effect of depleting NF- κ B in hypoxia. We don't show differential analysis comparing Rel depleted cells in hypoxia with control cells in hypoxia as we don't directly use that information when defining NF- κ B-dependent hypoxia responsive genes. In Figure 2 we define NF- κ B-dependent hypoxia responsive genes, and, as previously suggested, we made a supplementary figure (Appendix Figure S2) to diagrammatically explain (with Venn diagrams) how we define NF- κ B-dependent hypoxia responsive genes. Genes which are upregulated/downregulated in siControl hypoxia compared to siControl normoxia but not in siRelA/siRelB/sicRel hypoxia compared to siControl normoxia are considered siRelA/siRelB/sicRel-dependent hypoxia responsive genes. The combination of siRelA/siRelB/sicRel-dependent hypoxia responsive genes are considered NF- κ B-dependent hypoxia responsive genes.

We have modified the text for Figure 2 and the text and legend for Appendix Figure S2 to further improve clarity. Also, we changed the labelling of the figure 1F-G and figure 2D-E for more clear illustration.

-Inclusion of appropriate controls:

In Figure 3, normoxic controls should be included for all models to ensure that reduced induction upon hypoxia is not simply a reflection of lower basal expression levels (similar to that mentioned for Figure 1). In other words, the normoxia/hypoxia comparison for each condition should be shown and supported by appropriate statistical analysis.

please see referee #2 comment 1 response.

-Quantification and statistical rigor:

The manuscript generally lacks quantitative data presentation. For example, in Figure S3, the identification of HIF1 binding motifs in RELA, RELB, or c-REL-dependent genes should be compared to randomly selected hypoxia-induced genes to assess enrichment significance.

We disagree with the statement that the manuscript generally lacks quantitative data presentation. We have performed the suggested analysis (Shown below). As expected, HIF subunit binding sites motifs are also enriched when looking at a randomly selected set of hypoxia upregulated genes. We added the following statement to the main text in order to clarify that enrichment of HIF motifs are not surprising; “This is not surprising given the dominant role of HIF transcription factors in hypoxia induced gene activation”.

Randomly-selected 24 h Hypoxia Up DEGs

Motif	Protein	P value
	HIF-1α	1X10 ⁻¹⁰
	HIF-1β	1X10 ⁻⁵
	HIF-2α	1X10 ⁻⁴
	Hoxa13	1X10 ⁻³
	TATA-Box	1X10 ⁻²
	Cdx2	1X10 ⁻²

In Figures 7A-C, the reductions in protein levels and their reversal upon NF-κB depletion are, at best, marginal. The sentence stating that "Hypoxia led to reductions in several proteins represented in this panel..." is only partially supported by the data (perhaps only for IDH1). Similarly, differences in Vimentin and DYRK levels are negligible. Quantification with individual replicate data points should be provided for all experiments.

As suggested by reviewer 2, we do provide quantification, and we have moved this to supplementary. We removed all statistics regarding Western blot quantification as well as these are not appropriate for Western blotting. We disagree with the reviewer here and do believe there are differences in the proteins we have analysed.

General comment:

Overall, the manuscript requires substantial revision of both text and figures to improve clarity and to present conclusions that are rigorously supported by the data. Including appropriate controls and clear normalization strategies is essential. At this stage, in my view, the manuscript does not provide sufficient experimental evidence to support several of the main claims. Additionally, the presentation makes it unnecessarily difficult for the general scientific community to distinguish between robust findings and observations that could be due to experimental variability. I recommend that the authors carefully revise the whole manuscript to include all necessary controls, improve figure clarity, provide detailed quantifications, and rephrase conclusions where needed to reflect the actual strength of the data presented.

We have carefully revised the manuscript based on the reviewer's suggestions.

Cross-comments from referee 1:

In principle, I agree with the reviewer on the importance of normoxic controls. However, the authors may not be in a position to repeat all of the high throughput experiments with normoxic stimuli as this could be inordinately expensive. In this case, the authors could confirm some of their key genes in isolation and explain clearly in the text the limitations of not including normoxic samples. I still believe the paper is worthy of publication.

Cross-comments from referee 2:

In my opinion this experiment is not essential but is the optimal way to address the question. The authors have already provided some evidence that not all genes affected by NF κ B in hypoxia are affected by silencing of NF κ B family members in normoxia which in my view is the main point. Doing further experiments is unlikely to change this main finding.

Cross-comments from referee 3:

I think it is reasonable to request validation of a panel of candidate NF- κ B-dependent hypoxia-responsive genes by qPCR, including a normoxia-hypoxia conditions in the different cellular models, as suggested by reviewer 2. I still stand by my other comments about over-interpretation and a lack of clarity in several of the experiments. Could you please ask the authors to revise the text in general to improve the connection between what is observed and what is expressed?

Response to cross-comments from referee 1,2 and 3

Please see referee #2 comment 1 response.

Prof. Sonia Rocha
University of Liverpool
Institute of Systems Molecular and Integrative Biology
Crown street
Liverpool, Merseyside L697ZB
United Kingdom

Dear Sonja,

I am very pleased to accept your manuscript for publication in the next available issue of EMBO reports. Thank you for your contribution to our journal.

Referee #3:

I do not have any further comments. In my opinion, the authors have now revised the manuscript in line with the reviewers' comments.
